

# A new basal ornithopod dinosaur from the Lower Cretaceous of China

Yuqing Yang[1,2,3], Wenhao Wu[4,5], Paul-Emile Dieudonné[6] and Pascal Godefroit[7]

[1] College of Resources and Civil Engineering, Northeastern University, Shenyang, Liaoning, China
[2] College of Paleontology, Shenyang Normal University, Shenyang, Liaoning, China
[3] Key Laboratory for Evolution of Past Life and Change of Environment, Province of Liaoning, Shenyang Normal University, Shenyang, Liaoning, China
[4] Key Laboratory for Evolution of Past Life and Environment in Northeast Asia, Ministry of Education, Jilin University, Changchun, Jilin, China
[5] Research Center of Paleontology and Stratigraphy, Jilin University, Changchun, Jilin, China
[6] Instituto de Investigación en Paleobiología y Geología, CONICET, Universidad Nacional de Río Negro, Rio Negro, Argentina
[7] Directorate 'Earth and History of Life', Royal Belgian Institute of Natural Sciences, Brussels, Belgium

Corresponding author
Pascal Godefroit,
Pascal.Godefroit@naturalsciences.be

## ABSTRACT

A new basal ornithopod dinosaur, based on two nearly complete articulated skeletons, is reported from the Lujiatun Beds (Yixian Fm, Lower Cretaceous) of western Liaoning Province (China). Some of the diagnostic features of *Changmiania liaoningensis* nov. gen., nov. sp. are tentatively interpreted as adaptations to a fossorial behavior, including: fused premaxillae; nasal laterally expanded, overhanging the maxilla; shortened neck formed by only six cervical vertebrae; neural spines of the sacral vertebrae completely fused together, forming a craniocaudally-elongated continuous bar; fused scapulocoracoid with prominent scapular spine; and paired ilia symmetrically inclined dorsomedially, partially covering the sacrum in dorsal view. A phylogenetic analysis places *Changmiania liaoningensis* as the most basal ornithopod dinosaur described so far. It is tentatively hypothesized that both *Changmiania liaoningensis* specimens were suddenly entrapped in a collapsed underground burrow while they were resting, which would explain their perfect lifelike postures and the complete absence of weathering and scavenging traces. However, further behavioural inference remains problematic, because those specimens lack extensive sedimentological and taphonomic data, as it is also the case for most specimens collected in the Lujiatun Beds so far.

## INTRODUCTION

The Lujiatun Beds of the Yixian Formation are the lowermost fossil-bearing horizon of the Jehol Group in western Liaoning Province, China (*He et al., 2006*). Thousands of perfectly preserved vertebrate fossils have been unearthed from this horizon, including lizards, mammals and dinosaurs. Unlike in other horizons from the Jehol Biota, the Lujiatun fossils are preserved in three dimensions, revealing unique behaviour information, such as parental care (*Meng et al., 2004*) and sleeping posture (*Xu & Norell, 2004*) of dinosaurs,

and mammals eating dinosaurs (*Hu et al., 2005*). Those fossils are preserved in 5–20-m-thick tuffs exposed close to the Lujiatun Village, Shanyuan, Beipiao City. $^{40}$Ar/$^{39}$Ar dating of ash from the Lujiatun beds interbedded with the fossiliferous layers shows that the Lujiatun specimens are Barremian (123.2 ± 1.0 Ma) in age (*He et al., 2006*). It has been hypothesized that the Lujiatun fauna was killed catastrophically by lahar from a nearby shield volcano (*He et al., 2006*; *Zhao, Barrett & Eberth, 2007*). A wide variety of theropod dinosaurs has been discovered in the Lujiatun Beds, including the tyrannosauroid *Dilong paradoxus* (*Xu et al., 2004*), the oviraptorosaur *Incisivosaurus gauthieri* (*Xu et al., 2002a*), the ornithomimosaurs *Shenzhousaurus orientalis* (*Ji et al., 2003*) and *Hexing qingyi* (*Jin, Chen & Godefroit, 2012*), the dromaeosaurid *Graciliraptor lujiatunensis* (*Xu & Wang, 2004a*), and the troodontids *Sinovenator changii* (*Xu et al., 2002b*), *Mei long* (*Xu & Norell, 2004*), *Sinusonasus magnodens* (*Xu & Wang, 2004b*), *Daliansaurus liaoningensis* (*Shen et al., 2017a*), and *Liaoningvenator curriei* (*Shen et al., 2017b*). Among ornithischians, psittacosaurs are extremely abundant, being represented by the species *Psittacosaurus lujiatunensis* (including *P. major* and *Hongshanosaurus houi*; *Brandon & Dodson, 2013*). The neoceratopsian *Liaoceratops yanzigouensis* (*Xu et al., 2002a*) and the ornithopod *Jeholosaurus shangyuensis* (*Xu, Wang & You, 2000*) complete the dinosaur fauna.

Here we describe a new ornithischian dinosaur from the Lujiatun Beds, represented by two nearly complete and articulated specimens housed in the Paleontological Museum of Liaoning (PMOL) in Shenyang. Although it superficially resembles *Jeholosaurus*, represented by numerous specimens in the same deposits, the observed differences suggest that this new taxon occupies a more basal phylogenetic position at the base of the clade Ornithopoda.

## MATERIALS AND METHODS

### Origin of the studied specimens

As it is the case for most of the dinosaur specimens known from western Liaoning Province, the holotype and referred specimens of *Changmiania liaoningensis* were acquired by the PMOL from local farmers, according to whom the specimens were collected in the Lujiatun Beds close to Lujiatun Village. The specimens were only partially prepared when they were acquired by the PMOL. PMOL AD00114 was subsequently carefully prepared by the PMOL technical staff. Careful examination by the authors of the present paper and X-ray analyses did not reveal any trace of forgery besides the usual restorations: some of the bones that have been slightly broken off or damaged during the preparation process have been glued together, and fragile fragments of bones have been cemented, but strictly following the original contour of the bone before restoration. The anatomy of the two specimens is fully concordant and clearly different from the *Jeholosaurus* specimens described from the same formation. Moreover, the morphological and phylogenetic information collected from the studied specimens is not discordant with our current understanding of basal ornithopod anatomy and evolution. Based on our close examination of the blocks and our previously accumulated rich experience with

Liaoning specimens, we can therefore confidently guarantee the authenticity of the specimens.

## Phylogenetic nomenclature

For consistency purpose, we have tried to comply as far as possible with the phylogenetic nomenclature adopted by *Butler, Upchurch & Norman (2008)* and *Boyd (2015)* in some of the most recent revisions of ornithischian phylogeny. We therefore use the following definitions of higher-level ornithischian taxa in the present paper:

- Cerapoda *Sereno (1986)*: *Parasaurolophus walkeri*, *Triceratops horridus*, their most recent common ancestor and all descendants (*Butler, Upchurch & Norman, 2008*);

- Clypeodonta *Norman (2015)*: *Hypsilophodon foxii*, *Edmontosaurus regalis*, their most recent common ancestor, and all of its descendants (*Norman, 2015*);

- Genasauria *Sereno (1986)*: *Ankylosaurus magniventris*, *Stegosaurus stenops*, *Parasaurolophus walkeri*, *Triceratops horridus*, *Pachycephalosaurus wyomingensis*, their most recent common ancestor and all descendants (*Butler, Upchurch & Norman, 2008*).

- Heterodontosauridae *Romer (1966)*: all ornithischians more closely related to *Heterodontosaurus tucki* than to *Parasaurolophus walkeri*, *Pachycephalosaurus wyomingensis*, *Triceratops horridus*, or *Ankylosaurus magniventris* (*Sereno, 2005*);

- Iguanodontia *Dollo (1888)*: all ornithopods more closely related to *Parasaurolophus walkeri* than to *Hypsilophodon foxii* or *Thescelosaurus neglectus* (*Sereno, 2005*);

- Jeholosaurinae *Han et al. (2012)*: all ornithopods more closely related to *Jeholosaurus shangyuanensis*, than to *Changmiania liaoningensis*, *Orodromeus makelai*, *Nanosaurus agilis*, *Hypsilophodon foxii*, or *Thescelosaurus neglectus* (modified from *Han et al., 2012*);

- Marginocephalia *Sereno (1986)*: *Triceratops horridus*, *Pachycephalosaurus wyomingensis*, their most recent common ancestor and all descendants (*Butler, Upchurch & Norman, 2008*);

- Neornithischia *Cooper (1985)*: all genasaurians more closely related to *Parasaurolophus walkeri* than to *Ankylosaurus magniventris* or *Stegosaurus stenops* (*Butler, Upchurch & Norman, 2008*);

- Ornithischia *Seeley (1887)*: all dinosaurs more closely related to *Triceratops horridus* than to *Passer domesticus* or *Saltasaurus loricatus* (*Butler, Upchurch & Norman, 2008*);

- Ornithopoda *Marsh (1881)*: all genasaurians more closely related to *Parasaurolophus walkeri* than to *Triceratops horridus* (*Butler, Upchurch & Norman, 2008*);

- Orodrominae *Brown et al. (2013)*: all ornithopods more closely related to *Orodromeus makelai* than to *Nanosaurus agilis* or *Thescelosaurus neglectus* (this study);

- Parksosaurinae *Buchholz (2002)*: all clypeodonts more closely related to *Parksosaurus warreni* than to *Thescelosaurus neglectus* or *Parasaurolophus walkeri* (*Boyd, 2015*, modified);

- Thyreophora *Nopcsa (1915)*: all genasaurians more closely related to *Ankylosaurus magniventris* than to *Parasaurolophus walkeri*, *Triceratops horridus*, or *Pachycephalosaurus wyomingensis* (*Butler, Upchurch & Norman, 2008*).

## Phylogenetic analysis

To assess its phylogenetic position within ornithischian dinosaurs, we included *Changmiania liaoningensis* in an extensively modified version of the character-taxon matrix published by *Dieudonné et al. (2016)*. Besides *Changmiania liaoningensis*, we have integrated 14 taxa that were not included in *Dieudonné et al. (2016)* data-matrix: Ankylosauria, *Archaeoceratops oshimai, Chaoyangsaurus youngi, Goyocephale lattimorei, Homalocephale calathocercos, Isaberrysaura mollensis, Kulindadromeus zabaikalicus, Laquintasaura venezuelae, Liaoceratops yanzigouensis, Morrosaurus antarcticus*, Stegosauria, *Stenopelix valdensis, Thescelosaurus assiniboiensis* and *Wannanosaurus yansiensis*. In the present analysis, we have not coded the basal ornithopod *Oryctodromeus cubicularis*, from the middle Cretaceous Blackleaf Formation of southwestern Montana and the Wayan Formation of southeastern Idaho, USA (*Varricchio, Martin & Katsura, 2007*; *Krumenacker, 2010*, *2017*), pending the formal publication of the numerous partial skeletons from the Wayan Formation (*Krumenacker, 2010*, *2017*). The final analysis consequently includes 61 taxonomic units and 263 characters (Table S1 provides the character descriptions, Table S2 contains the final data matrix, Table S3, the data matrix in TNT format, and Table S4, the character support for selected nodes). *Herrerasaurus ischigualastensis* was used as outgroup. The analysis of the dataset was run using TNT (Tree Analysis using New Technology, *Goloboff, Farris & Nixon (2008)*); all the characters were equally weighted and regarded as non-additive. Several characters were coded as additive (#110, #149, #152, #205, #208) in the original matrix published by *Dieudonné et al. (2016)*; however, we estimate that assessing whether a character is additive is in most cases rather subjective and, by experience, subject to discussions, depending on the final result we have in mind before running the analysis; to avoid any circular reasoning, we have therefore decided to regard all characters as non-additive in the present analysis. We performed a first round of 100 'New Technology' search analyses, with default parameter, and then explored the shortest tree islands found performing Tree Bisection Reconnection analyses saving all shortest tree found. Bremer nodal support was calculated in TNT saving all trees up to 10 steps longer than the most parsimonious results.

## Nomenclatural acts

The electronic version of this article in Portable Document Format will represent a published work according to the International Commission on Zoological Nomenclature (ICZN), and hence the new names contained in the electronic version are effectively published under that Code from the electronic edition alone. This published work and the nomenclatural acts it contains have been registered in ZooBank, the online registration system for the ICZN. The ZooBank Life Science Identifiers (LSIDs) can be resolved and the associated information viewed through any standard web browser by appending the LSID to the prefix http://zoobank.org/. The LSID for this publication is: urn:lsid:zoobank.org:pub:9C5F2451-4E00-4919-9FE9-14E6629FCF64. The online version of this work is archived and available from the following digital repositories: PeerJ, PubMed Central and CLOCKSS.

# SYSTEMATIC PALEONTOLOGY

Dinosauria *Owen (1842)*
Ornithischia *Seeley (1887)*
Neornithischia *Cooper (1985)*
Ornithopoda *Marsh (1881)*
*Changmiania liaoningensis* nov. gen., nov. sp.

Etymology — *Changmian*: eternal sleep, in Chinese Pinyin; *liaoningensis*: from Liaoning.

Holotype — PMOL AD00114, a nearly complete articulated skeleton (Figs. 1A and 1B)

Referred specimen — PMOL LFV022, another nearly complete skeleton, in dorsal view (Fig. 1C).

Locality and Horizon — Lujiatun, Shangyuan, Beipiao City, western Liaoning, China; Lujiatun Beds, lowest beds of Yixian Formation, Barremian, Lower Cretaceous.

Diagnosis (autapomorphies preceded by an asterisk): Fused premaxillae; nasal laterally expanded, overhanging the maxilla; long and straight posterior process of supraorbital, nearly contacting the postorbital; *rostrocaudally elongated frontals: maximum length to width ratio >4; *no sagittal crest on parietal; infratemporal fenestra triangular in lateral view, wider dorsally than ventrally; jugal process of postorbital rostrocaudally narrow along its whole height; *rostral process of squamosal straight and rostrocaudally elongated; *prominent caudal boss at the dorsolateral corner of the squamosal; *ventral border of dentary convex and ventral border of angular deeply concave, so that the lower jaw looks sigmoidal in lateral view; six cervical vertebrae; *neural spines of the sacral vertebrae completely fused together, forming a craniocaudally-elongated continuous bar; fused scapulocoracoid; *scapular blade asymmetrically expanded both ventrally and dorsally; *paired ilia symmetrically inclined dorsomedially, partially covering the sacrum in dorsal view; dorsal margin of ilium regularly convex along the whole length of the bone; *proximal half of fibula as robust as the corresponding portion of the tibia.

## Description

The osteological description of *Changmiania liaoningensis* proposed below is mainly based on its holotype PMOL AD00114, except when variation can be observed with referred specimen PMOL LFV022 or when some parts of the skeleton (e.g. the hands) are better exposed in the latter specimen. References to PMOL LFV022 are in those cases explicitly indicated in the text.

### Skull

Premaxilla — The maxillary process of the premaxilla is particularly robust, trapezoidal in shape, and inclined both caudodorsally and dorsolaterally (Fig. 2). It forms the entire caudal margin of the external naris. It regularly expands rostrocaudally as it extends dorsally. Its caudal border contacts the maxilla and its straight dorsal border, the nasal. Its caudodorsal corner contacts the lacrimal, excluding the maxilla from the nasal, as also

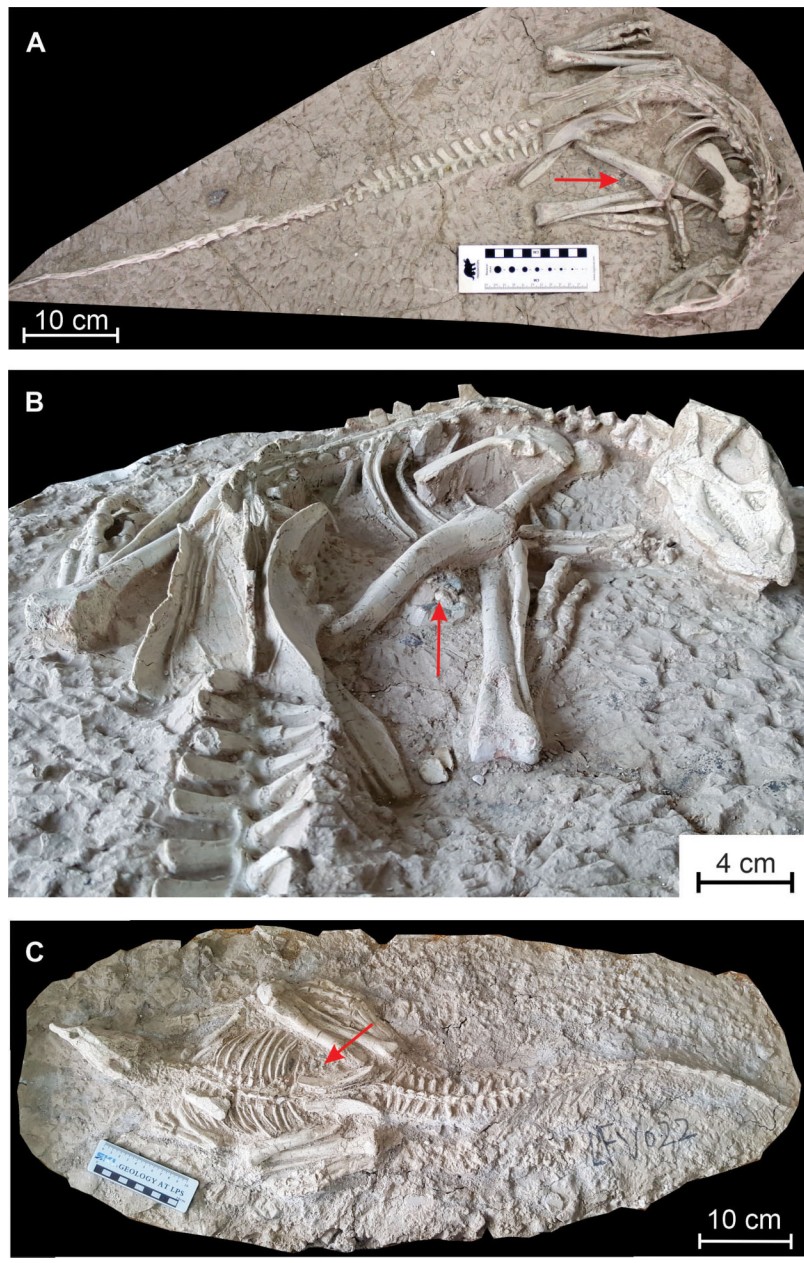

**Figure 1** *Changmiania liaoningensis*, **an ornithopod dinosaur from the Lower Cretaceous of Lujiatun (Liaoning Province, China).** (A) Holotype PMOL AD00114 in dorsal view; (B) anterior part of the holotype PMOL AD00114 in caudolateral view; (C) referred specimen PMOL LFV022 in dorsal view. Red arrows indicate the emplacement of the gastrolith clusters.

observed in *Heterodontosaurus* (*Sereno, 2012*), *Kulindadromeus* (*Godefroit et al., 2014*) and larger *Jeholosaurus* specimens (*Barrett & Han, 2009*). The much smaller nasal process forms the rostrodorsal border of the external naris; as also observed in *Jeholosaurus* (*Barrett & Han, 2009*), it only participates in the rostral third of the narial dorsal margin (Figs. 2). This process rapidly tapers both mediolaterally and dorsoventrally as it extends dorsally, terminating in a fine point, so that the external naris is highly constricted in

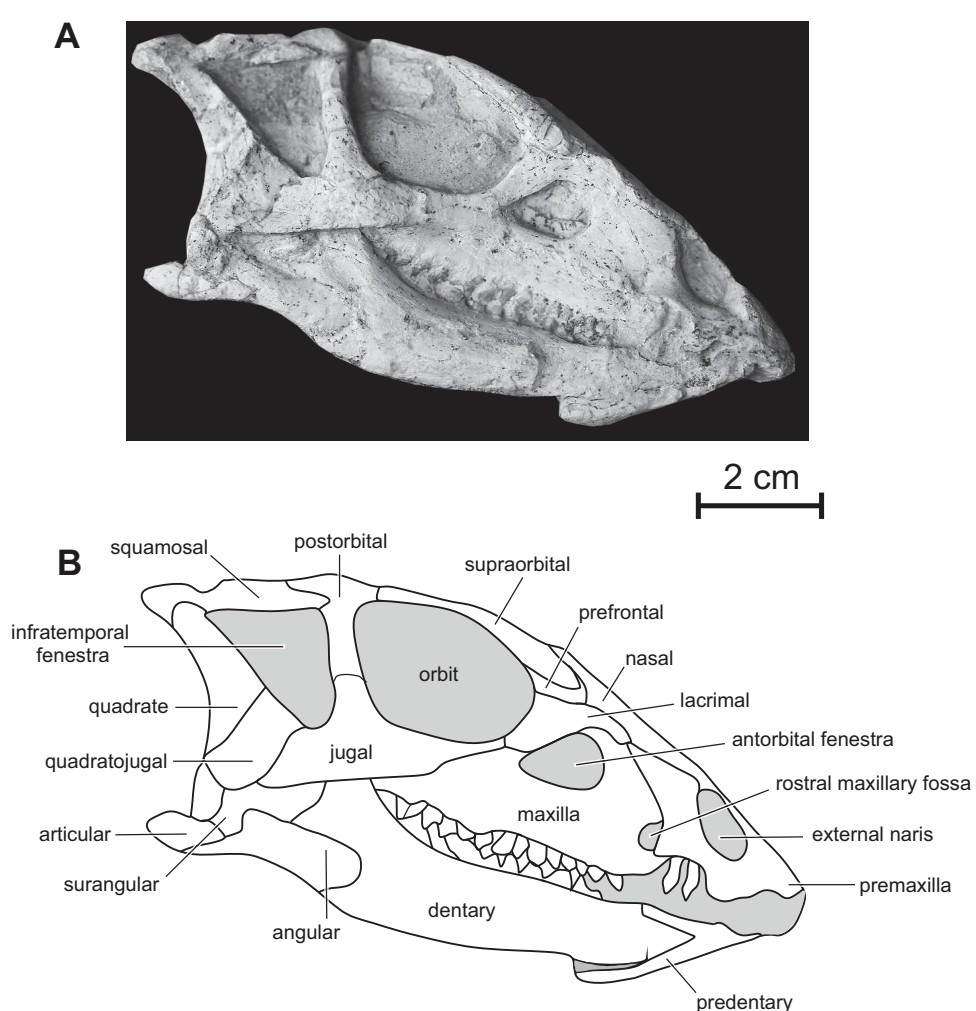

**Figure 2 Skull of PMOL AD00114 in right lateral view.** (A) Photograph; (B) line drawing.

dorsal view, at the level of the contact between the paired premaxillae and nasals (Figs. 3 and 4). The rostral end of the nasal slightly overlaps the caudal end of the premaxilla. In dorsal view, the premaxillae are nearly completely fused together; only the distal end of their nasal process is not fused. Extensive fusion of the premaxillae has been reported in *Changchunsaurus* (*Jin et al., 2010*), *Oryctodromeus* and *Zephyrosaurus*, and interpreted as an adaptation to burrowing behavior (*Varricchio, Martin & Katsura, 2007*). The rostral surface of the paired premaxillae is rugose (Fig. 5A), as also observed in *Hypsilophodon* (*Galton, 1974*), *Zephyrosaurus* (*Sues, 1980*), *Jeholosaurus* (*Barrett & Han, 2009*) and *Changchunsaurus* (*Jin et al., 2010*), and likely served for the attachment of a rhamphotheca. The rugosities slightly overhang the rostroventral corner of the external naris. They extend along the rostroventral surface of the main body of the premaxilla and potentially obscure the presence of premaxillary foramina. Above those rugosities the lateral surface of the premaxilla is convex and forms a shelf, so that the external naris is strongly inset from the lateral surface of the premaxilla (Fig. 5A). It contrasts with the

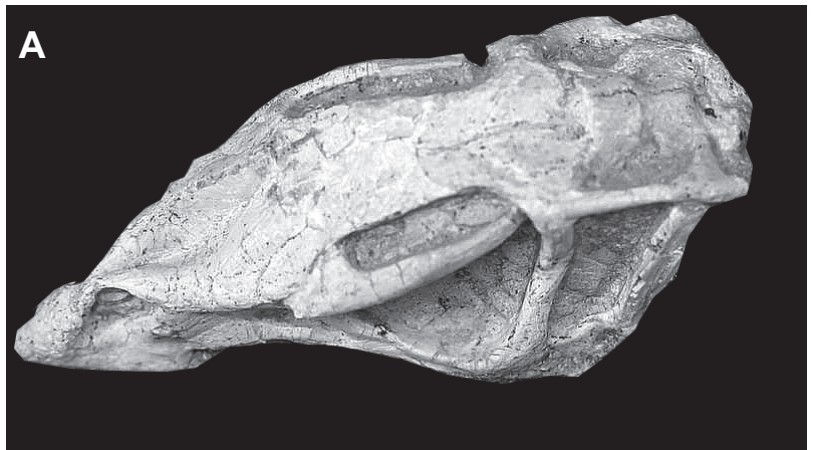

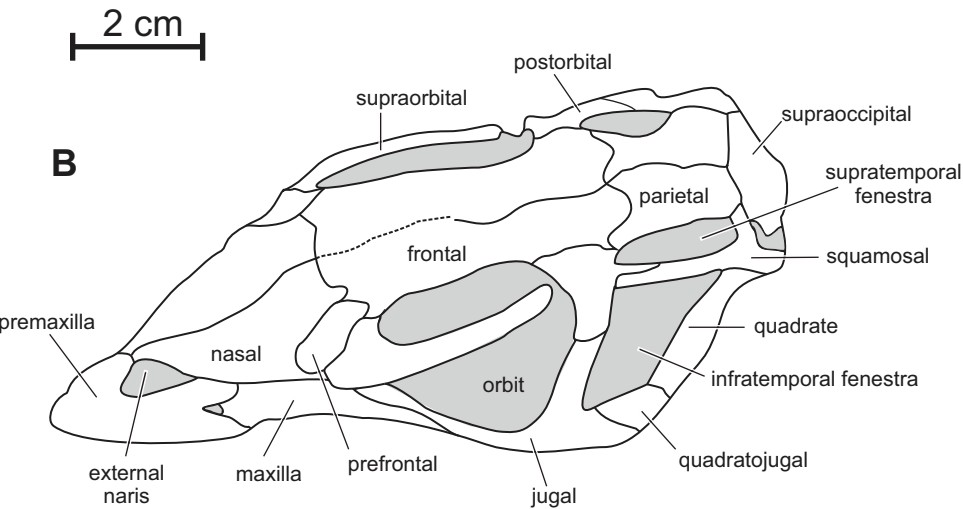

**Figure 3 Skull of PMOL AD00114 in left dorsolateral view.** (A) Photograph; (B) line drawing.

concave lateral surface of the premaxilla in *Jeholosaurus* (*Barrett & Han, 2009*) and *Changchunsaurus* (*Jin et al., 2010*). There is no distinct external narial fossa, as observed in some *Jeholosaurus* specimens (*Barrett & Han, 2009*). The lateroventral border of the premaxilla lies as about the same level as the ventral margin of the maxilla (Fig. 2), as also observed in *Zephyrosaurus* (*Sues, 1980*, fig. 16), *Orodromeus* (*Scheetz, 1999*), *Jeholosaurus* (*Barrett & Han, 2009*), *Changchunsaurus* (*Jin et al., 2010*) and *Haya* (*Makovicky et al., 2011*).

The number of premaxillary teeth cannot be precisely evaluated. Nor is it possible to determine if the first premaxillary tooth was located close to the rostral end of the premaxilla or if the rostral end of the premaxilla was edentulous. The premaxillary teeth are robust, with a well-marked constriction between the root and the crown (Fig. 5A). The crowns are clearly separated from each other as in other basal cerapodans, including *Hypsilophodon* (*Galton, 1974*), *Yinlong* (*Xu et al., 2006*), *Jeholosaurus* (*Barrett & Han, 2009*), and *Changchunsaurus* (*Jin et al., 2010*). The crowns have a subconical shape, with

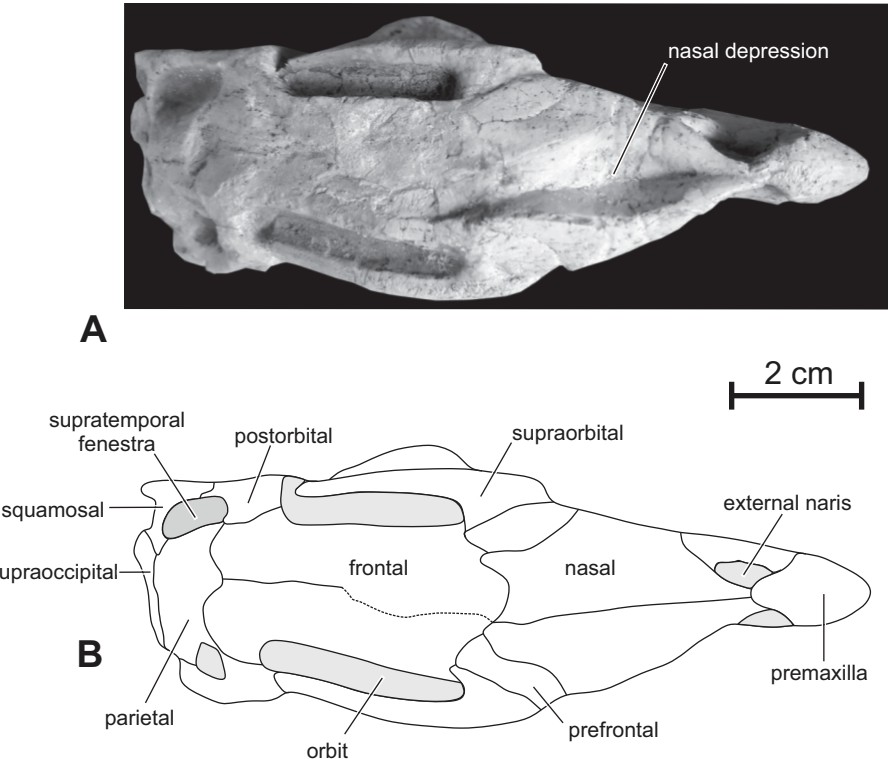

**Figure 4** **Skull of PMOL AD00114 in dorsal view.** (A) Photograph; (B) line drawing.

bulbous crown bases; they taper apically and are slightly recurved. They do not bear denticles, and their enamel is perfectly smooth.

**Maxilla** — On the rostral third of the maxilla, the ascending process is high and hook-like (Figs. 2 and 5B); its concave caudal border forms the rostral margin of the antorbital fenestra, whereas its straight rostral border contacts the premaxilla. The apex of the ascending process contacts the lacrimal. A small circular fossa is set rostrally at the base of the ascending process, at the junction with the premaxilla (Figs. 2 and 5B). A similar fossa is present in *Hypsilophodon* (*Galton, 1974*; NHMUK R197), *Jeholosaurus* (*Xu, Wang & You, 2000*), *Changchunsaurus* (*Jin et al., 2010*), *Haya* (*Makovicky et al., 2011*) and *Orodromeus* (*Scheetz, 1999*), and was identified as a synapomorphy of Ornithopoda by *Butler, Upchurch & Norman (2008)*. The tooth-bearing ramus is rostrocaudally elongated and maintains an approximately even height along most of its length (Fig. 5B). A horizontal ridge extends rostrally from the ventral margin of the jugal and limits a wide and dorsoventrally concave buccal emargination (Figs. 5B and 5C). The distance between the horizontal ridge and the ventral margin of the external antorbital fenestra is significantly less than in *Changchunsaurus* (*Jin et al., 2010*), reflecting the larger size of the external antorbital fenestra. A horizontal series of small nutrient formina is present ventrally to the horizontal ridge (Fig. 5B). The dorsal portion of the medial lamina of the maxilla is visible within the antorbital fossa (Fig. 5B).

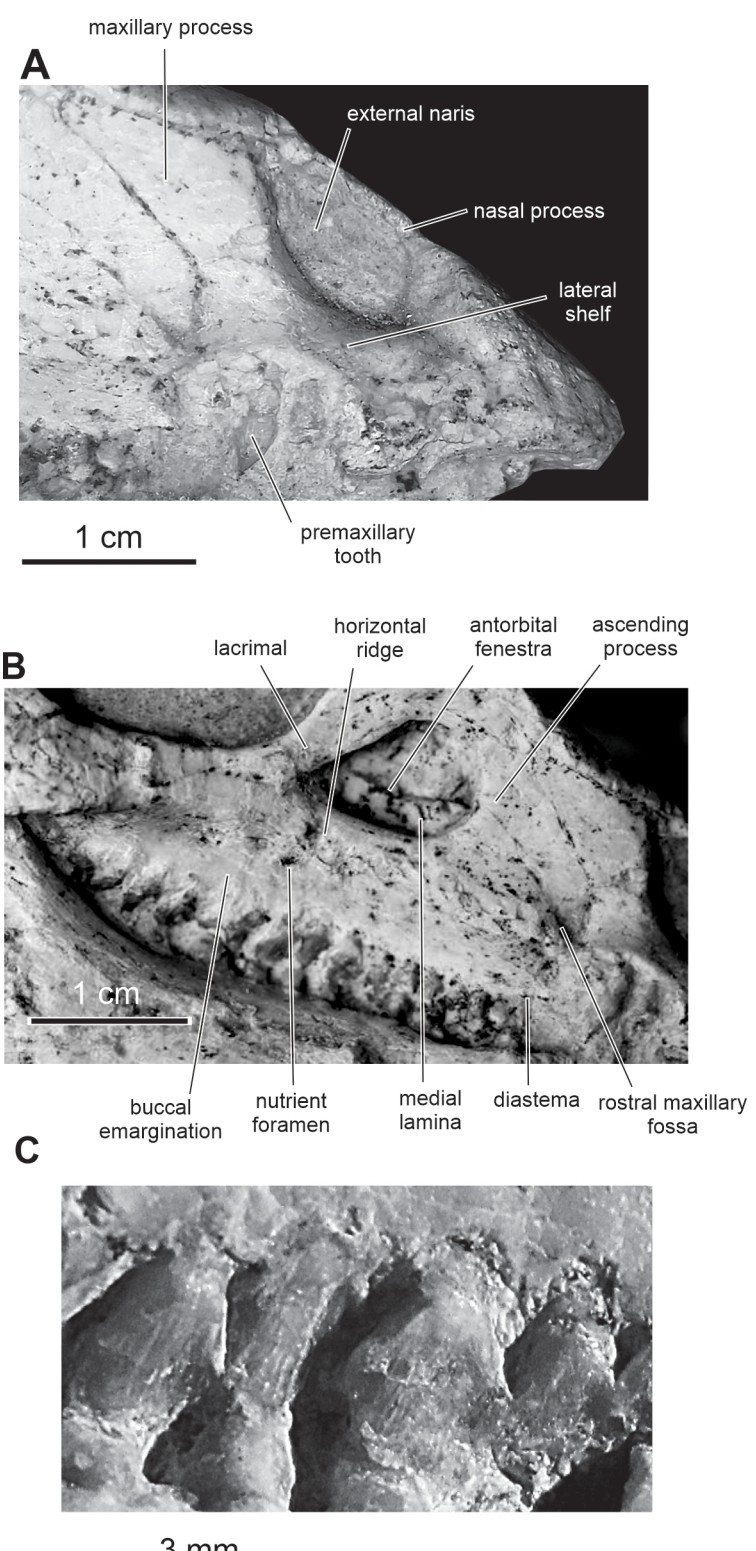

**Figure 5 Skull of PMOL AD00114 in right lateral view.** (A) Close-up of the premaxillary region; (B) close-up of the maxillary region; (C) close-up of the maxillary teeth.

Fifteen maxillary teeth can be observed on the right maxilla of the holotype. A short edentulous diastema, equivalent to the mesiodistal length of 1 or 2 crowns, is present at the rostral end of the maxilla (Fig. 5B), as in *Changchunsaurus* (*Jin et al., 2010*). The size of the maxillary crowns progressively increases up to the mid-length of the maxilla. The crowns are poorly preserved and/or partly destroyed during preparation. As also observed in *Changchunsaurus* (*Jin et al., 2010*), the maxillary crowns are imbricated, the distal part of each crown overlaping laterally the mesial part of the succeeding one (Fig. 5C). The crowns are as high as mesiodistally long. Small denticles, supported by ridges extending from a weak basal cingulum, are developed along the apical half of the crowns; those ridges look better developed than in *Jeholosaurus* (*Barrett & Han, 2009*) and *Changchunsaurus* (*Jin et al., 2010*), more closely resembling the condition in *Hypsilophodon* (*Galton, 1974*). The maxillary crowns of *Orodromeus* apparently lack ridges supporting their marginal denticles (*Scheetz, 1999*). A low median eminence is usually developed on the labial surface, but a prominent primary ridge is absent.

**Lacrimal** — As is usual in basal cerapodans, the lacrimal consists of a ventral and dorsal process forming an inverted 'L' in lateral view (Figs. 2 and 6A). Both processes form an angle of approximately 120 degrees, as in *Jeholosaurus*. The ventral process becomes more rostrocaudally restricted ventrally. This process separates the rostral margin of the orbit from the caudal margin of the antorbital fossa; it contacts ventrally the maxilla and the jugal. The dorsal process extends rostrally to contact the maxillary process of the premaxilla. Its dorsal border contacts the prefrontal and nasal; ventrally it participates in the apical margin of the antorbital fossa and it contacts the ascending process of the maxilla (Fig. 6A).

**Nasal** — In dorsal view, the paired nasals taper rostrally above the external nares, where they contact the fused premaxillae to form the internarial bar; they quickly expand laterally and overhang the maxilla (Fig. 3) as it is also observed, to a lesser degree, in *Haya* (see *Norell & Barta, 2016*, fig. 9). At this level, their lateral margin is particularly thickened. There is obviously no row of foramina along the lateral aspect of the nasal, as observed in *Jeholosaurus* (*Xu, Wang & You, 2000*; *Barrett & Han, 2009*). The caudolateral border of the nasal is notched by the contact facet for the prefrontal. Its caudal margin forms a concave articular surface for the frontal (Fig. 3). The dorsal surface of the paired nasals forms a deep longitudinal depression (Fig. 4), present in a number of basal neornithischians (*Barrett, Butler & Knoll, 2005*), basal ornithopods, and ceratopsians, but which appears better developed than in other taxa in which this structure was described so far, including *Jeholosaurus* (*Xu, Wang & You, 2000*; *Barrett & Han, 2009*), *Haya* (*Makovicky et al., 2011*), *Liaoceratops* (*Xu et al., 2002a*) and *Yinlong* (*Xu et al., 2006*).

**Prefrontal** — the prefrontal is a strap-like element that participates in the rostrodorsal margin of the orbit. It articulates with the lacrimal ventrally, the nasal medially and overlaps the frontal caudally (Figs. 3 and 4). Its straight lateral border forms an extended articulation surface with the palpebral. There is no ventral process descending from the

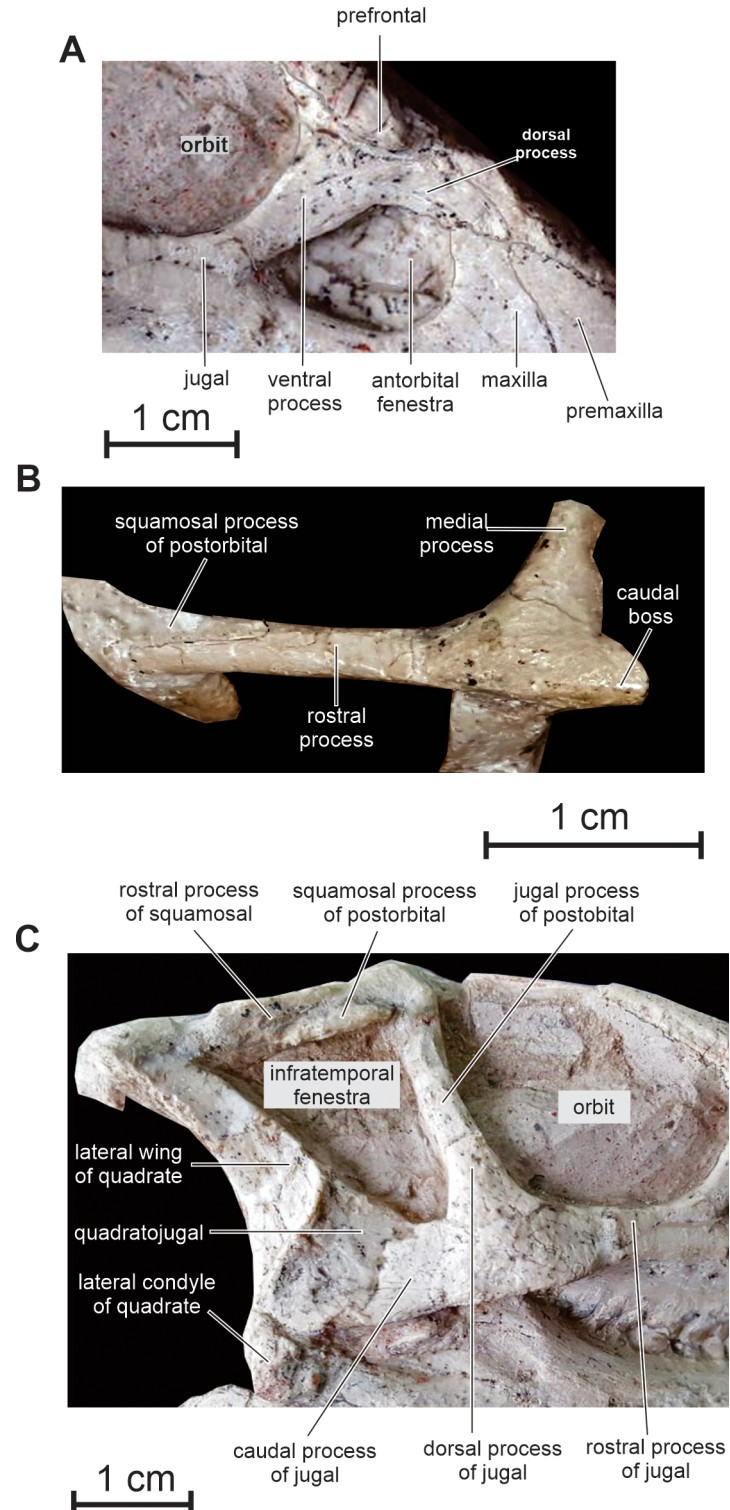

**Figure 6 Skull of PMOL AD00114.** (A) Antorbital fenestra region in right lateral view; (B) left squamosal and postorbital in dorsolateral view; (C) caudal half of the skull in right lateral view.

rostral end of the prefrontal, as observed in *Orodromeus* (*Scheetz, 1999*) and *Haya* (*Makovicky et al., 2011*).

**Supraorbital —** The rostral end of the supraorbital forms a medial and a rostral process that both articulate with the lateral edge of the prefrontal. Its caudal process is robust, straight and particularly, long spanning over nearly the entire diameter of the orbit (Fig. 4). However, it likely did not articulate with the postorbital as in *Agilisaurus* (*Peng, 1992*; *Barrett, Butler & Knoll, 2005*). There is no trace of a postpalpebral in articulation with the postorbital, as observed in *Thescelosaurus* (*Boyd, 2014*) and *Haya* (*Makovicky et al., 2011*). Together with the nasals and the prefrontals, the supraorbitals overhang the antorbital and orbital regions of the skull and likely protected those sensitive areas in life.

**Frontal —** In dorsal view, the frontal is subrectangular and slightly convex rostrocaudally. It is particularly rostrocaudally elongated, more than two times as long the parietal, and mediolaterally narrow: its maximum length to width ratio is >4 (Fig. 4). It therefore contrasts with the distinctly wider frontal in other basal ornithichians, ornithopods, and ceratopsians: *Jeholosaurus* (3: *Barrett & Han, 2009*), *Agilisaurus* (3.0: *Peng, 1992*), *Hexinlusaurus* (2,2: *He & Cai, 1984*), *Hypsilophodon* (3.2: *Galton, 1974*), *Zephyrosaurus* (3.0: *Sues, 1980*), *Orodromeus* (2.2: *Scheetz, 1999*), *Thescelosaurus* (1.9, *Galton, 1997*), *Liaoceratops* (2.2: *Xu et al., 2002a*), *Psittacosaurus* (1.8 : *Sereno et al., 1988*) and *Yinlong* (1.8: *Xu et al., 2006*). Each frontal forms about three quarters of the dorsal margin of the orbit, and the orbital rim is thin and rugose, as is typical of basal ornithopods (*Norman et al., 2004*; *Makovicky et al., 2011*); in *Jeholosaurus* (*Barrett & Han, 2009*) and *Haya* (*Makovicky et al., 2011*), the frontal comprises approximately 50% of the dorsal orbital margin. Rostrally, the paired frontals wedge between the nasals and are slightly overlapped by the prefrontals along their orbital margins. Caudolaterally, they form a straight contact with the postorbital and their caudal border is notched by the rostral process of the parietal. The frontals are excluded from the margin of the supratemporal fenestra by a direct contact between the postorbital and parietal (Figs. 3 and 4).

**Parietal —** The parietal is wide and robust, and its dorsal surface is regularly convex without any trace of a sagittal crest (Figs. 3 and 4), unlike in *Jeholosaurus* (*Barrett & Han, 2009*), *Haya* (*Makovicky et al., 2011*), *Hypsilophodon* (*Galton, 1974*) and *Zephyrosaurus* (*Sues, 1980*). Rostrally, the parietals form a rounded process that wedges between the caudal margins of the frontals; in *Jeholosaurus*, on the contrary, the parietals are notched at the midline to receive a short triangular process from the caudal margin of the frontals (*Barrett & Han, 2009*). Lateral to this point, the margin of the parietal produces a sinuous articulation with the frontals in dorsal view. The paired parietals are slightly constricted in dorsal view at the level of the middle part of the supratemporal fenestra. The rostrolateral corner of the parietal flares laterally. The caudolateral corner of the parietal also extends laterally to contact the squamosal. The caudal border of the parietals forms a straight articulation with the supraoccipital. In *Jeholosaurus* (*Barrett & Han, 2009*) and *Haya* (*Makovicky et al., 2011*), the caudal edge of the parietal is deeply notched at the midline. There is no trace of a nuchal crest in *Changmiania*.

**Supraoccipital** — The surface of the supraoccipital is highly eroded in both the holotype and referred specimen. Moreover, the occipital region of both specimens is still embedded in the sediments. The supraoccipital is inserted between the lateral wings of the caudal end of the parietal and has a rhomboidal outline. It is steeply inclined rostrodorsally.

**Postorbital** — In lateral view, the postorbital has the shape of an inverted 'L' and is formed by a jugal and a squamosal processes, meeting at an angle of about 90 degrees (Figs. 2 and 6B). Unlike other basal neoceratopsians, ornithopods and ceratopsians, the postorbital is not rostrocaudally widened at the level of the junction between both processes, so that it does not look subtriangular in lateral view; consequently, the dorsal portion of the infratemporal fenestra is rostrocaudally wider than its ventral portion, so that the infratemporal fenestra is triangular in shape (Figs. 2 and 6B), contrasting with the elliptical infratemporal fenestra, with a dorsoventral long axis and a reduced dorsal margin, in other basal neoceratopsians, ornithopods and ceratopsians described so far. The postorbital and infratemporal fenestra more closely resemble the condition in *Heterodontosaurus* (*Charig & Crompton, 1974*). The jugal process is straight, slightly inclined rostrocaudally, and, as already described, its rostrocaudal length remains equal along its whole height. It forms the dorsal half of the caudal margin of the orbit and rostral margin of the infratemporal fenestra; ventrally, it overlaps the dorsal process of the jugal. The squamosal process is quite short and tapers caudally (Fig. 6B); it is overlapped by the rostral process of the squamosal. In dorsal view, the postorbital forms a short medial process that contacts the frontal medially, and the parietal caudomedially. As in *Changchunsaurus*, there is no trace of rugosities on the lateral surface of the postorbital. Nodular ornamentation is present on the postorbital in *Jeholosaurus* (*Barrett & Han, 2009*), and in some basal ceratopsians (e.g. *Archaeoceratops*: IVPP V11114) and *Yinlong* (*Xu et al., 2006*). In *Haya* (*Makovicky et al., 2011*), *Zephyrosaurus* (*Sues, 1980*) and *Orodromeus* (*Scheetz, 1999*), a rugose crest, which could have served as an area of attachment for the postpalpebral, projects from the orbital rim near the juncture of the ventral and medial rami, (*Makovicky et al., 2011*).

**Squamosal** — The squamosal participates in the caudal and lateral margins of the supratemporal fenestra, and in the dorsocaudal margin of the infratemporal fenestra. The rostral process is particularly long and perfectly straight (Figs. 2 and 6B), as a consequence of the important rostrocaudal elongation of the dorsal margin of the infratemporal fenestra; it largely overlaps the squamosal process of the postorbital (Fig. 6B). The medial process is robust and triangular in dorsal view; its dorsal surface is inclined caudoventrally. The dorsolateral corner of the squamosal forms a prominent caudal boss, which forms the dorsocaudal corner of the skull in lateral view (Fig. 6B). The quadrate cotyle is cup-shaped; the pre-and postquadratic processes are not preserved.

**Quadrate** — The quadrate closely resembles that of *Changchunsaurus* (*Jin et al., 2010*): its dorsal portion is distinctly curved caudally, so that its articulation with the squamosal is located caudal to the level of the ventral quadrate condyles (Figs. 2 and 6C). The lateral wing of the quadrate is wide and concave rostrocaudally; its rostral border is sigmoidal

in lateral view and participates in the caudodorsal margin of the infratemporal fenestra (Fig. 6C); ventrally, it is particularly thickened along its long contact area with the quadratojugal. Ventrally, the lateral condyle is partially exposed on the right quadrate; it is distinctly offset below the level of the maxillary tooth row, as observed in virtually all ornithischians, including basal forms such as *Lesothosaurus* (*Norman, Witmer & Weishampel, 2004*; *Jin et al., 2010*).

**Quadratojugal** — The quadratojugal is trapezoidal in lateral view and mediolaterally compressed (Figs.2 and 6C); it participates in the caudoventral margin of the infratemporal fenestra. Its caudal portion extensively overlaps the rostroventral margin of the quadrate, but it does not reach the ventral process of the squamosal, unlike in *Heterodontosaurus* (*Crompton & Charig, 1962*), *Lesothosaurus* (*Sereno, 1991*), and the basal iguanodontians *Dryosaurus* and *Dysalotosaurus* (*Norman, 2004*). The rostral portion of the quadratojugal is overlapped by the caudal ramus of the jugal. As in *Changchunsaurus* (*Jin et al., 2010*), *Parksosaurus* (*Galton, 1973*), and some specimens of *Orodromeus* (*Scheetz, 1999*), there is no trace of a quadratojugal foramen; a circular foramen pierces the lateral surface of the quadratojugal in *Hypsilophodon* (*Galton, 1974*), *Jeholosaurus* (*Xu, Wang & You, 2000*), *Haya* (*Makovicky et al., 2011*) and *Tenontosaurus* (*Winkler, Murry & Jacobs, 1997*). Caudoventrally, the quadratojugal extends above the quadrate condyles, as also observed in most basal cerapodans, including *Hypsilophodon* (*Galton, 1974*), *Jeholosaurus* (*Barrett & Han, 2009*), *Changchunsaurus* (*Jin et al., 2010*), *Haya* (*Makovicky et al., 2011*), *Orodromeus* (*Scheetz, 1999*), *Psittacosaurus* (*Sereno et al., 1988*) and *Yinlong* (*Xu et al., 2006*). In advanced ornithopods, the dorsoventral extent of the quadratojugal is reduced and the ventral margin does not approach the quadrate condyles (*Norman, 2004*).

**Jugal** — As in *Jeholosaurus* (*Barrett & Han, 2009*), *Changchunsaurus* (*Jin et al., 2010*), and basal neoceratopsians (*You & Dodson, 2003*; *Xu et al., 2002a*, *2006*), the jugal is bowed outwards quite strongly along its length (Fig. 3). As in *Haya* (*Makovicky et al., 2011*), the lateral surface of the jugal lacks the ornamentation or jugal bosses seen in *Jeholosaurus* (*Barrett & Han, 2009*), *Changchunsaurus* (*Jin et al., 2010*), *Orodromeus* (*Scheetz, 1999*) and *Zephyrosaurus* (*Sues, 1980*). The rostral process of the jugal regularly tapers rostrally; it forms the ventral margin of the orbit and contacts the maxilla and the lacrimal (Fig. 6C). The dorsal process of the jugal is slightly inclined caudodorsally. It is laterally overlapped by the jugal process of the postorbital and participates in the ventral halves of the rostral margin of the infratemporal fenestra and of the caudal margin of the orbit. The caudal process is rather short, dorsoventrally high and mediolaterally compressed. It forms the ventral margin of the infratemporal fenestra. Its suture with the quadratojugal is rather complex: ventrally, it forms a long and slender projection extending below the ventral edge of the quadratojugal (Fig. 6C). There is no dorsocaudal projection, so that the caudal end of the jugal does not appear bifid as in *Jeholosaurus* (*Barrett & Han, 2009*), *Haya* (*Makovicky et al., 2011*), *Thescelosaurus* (*Boyd, 2014*) and possibly *Changchunsaurus* (*Jin et al., 2010*).

**Lower jaw**

**Predentary** — The predentary is partly eroded and only visible in right lateral view (Fig. 2). It is an elongate arrow-shaped element that tapers rostrally to a sharp, slightly upturned, point, as also observed in *Jeholosaurus* (*Xu, Wang & You, 2000*), *Changchunsaurus* (*Jin et al., 2010*) and *Archaeoceratops* (*You & Dodson, 2003*). Along the lateral side of the predentary, a horizontal sulcus separates the rostrolateral and the rostroventral processes. The caudoventral process is particularly elongated and slightly bowed rostroventrally, extending along the rostroventral border of the dentary.

**Dentary** — The dentary is elongated and rather robust, forming about 56% of the length of the mandible (Fig. 2). Its ventral margin is regularly convex in lateral view, contrasting with the straighter ventral margin in *Hypsilophodon* (*Galton, 1974*), *Jeholosaurus* (*Barrett & Han, 2009*), *Changchunsaurus* (*Jin et al., 2010*) and *Haya* (*Makovicky et al., 2011*; *Norell & Barta, 2016*). Its dorsal and ventral margins remain subparallel along most of their length. The rostral end of the dentary is not slightly downturned as in *Changchunsaurus* (*Jin et al., 2010*). The lateral side of the dentary is dorsoventrally convex; the tooth row is strongly inset and is separated from lateral side of the dentary by a well-developed buccal platform, limited ventrally by a low ridge. There is no trace of a rugose thickening of the rostrodorsal margin of the dentary and of a horizontal groove extending ventral and adjacent to the tooth row, as observed in *Changchunsaurus* (*Jin et al., 2010*). The height of the buccal emargination is constant. The lateral surface of the dentary is irregularly pierced by small foramina. The number of dentary teeth cannot be ascertained in the holotype, because they are obscured by the overlying maxillae. An edentulous diastema was likely developed along the rostral portion of the dentary, as in *Changchunsaurus* (*Jin et al., 2010*). Caudally, the dentary contributes to the rostral half of a high and stout coronoid process that slopes caudodorsally. The position of the last dentary teeth relatively to the coronoid process cannot be observed. The caudal border of the dentary is dorsoventrally concave and overlaps the lateral surfaces of the angular and surangular. At mid-height, it forms a large triangular spur that extends caudally between the angular and surangular (Fig. 7). This spur is slightly developed in *Hypsilophodon* (*Galton, 1974*), *Jeholosaurus* (*Barrett & Han, 2009*), *Haya* (*Makovicky et al., 2011*), but is absent in *Changchunsaurus* (*Jin et al., 2010*) and *Thescelosaurus* (*Boyd, 2014*). The dentary forms a triangular rostral process, which is limited dorsally and ventrally by articular facets (Fig. 2): the dorsal facet articulates with the caudolateral process of the predentary and faces rostrally and slightly medially, whereas the ventral facet that contacts the caudoventral process of the predentary is much longer and faces ventrolaterally.

The dentary teeth are present in their sockets, but they are obscured by the sediment or the overlying maxillary teeth. Therefore, they cannot adequately be described.

**Surangular** — The right surangular is partly exposed in lateral view, but a large portion is obscured by the jugal and quadratojugal (Figs. 2 and 7). The surangular forms the caudal

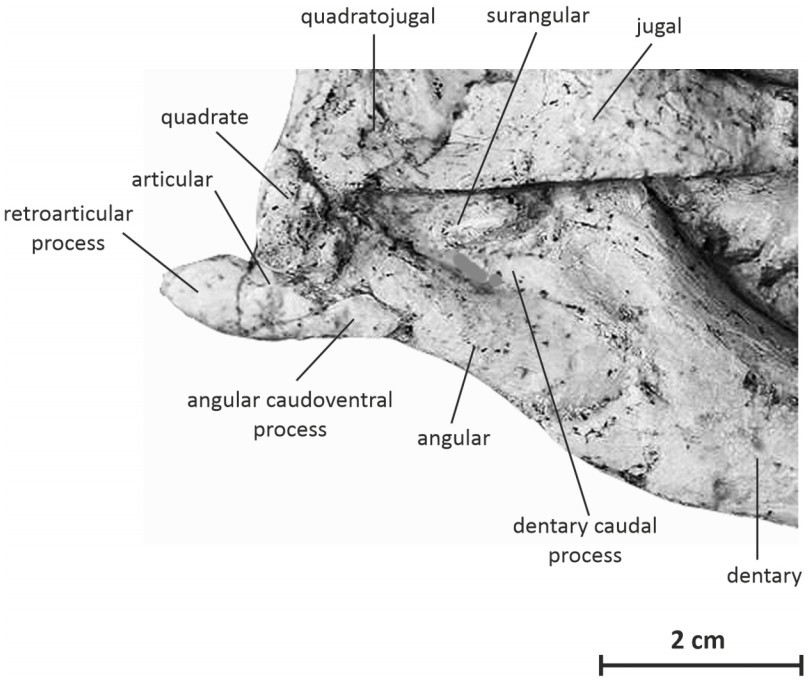

**Figure 7 Postdentary bones of JMOL AD00114 in right lateral view.**

half of the coronoid process. Its lateral margin forms a wide concave sulcus, close to the junction with the dentary and angular.

**Angular** — The angular is a rostrocaudally-elongated element that forms the posteroventral portion of the mandible (Figs. 2 and 7). It contacts the dentary rostrally and is overlapped dorsally by the surangular. Its lateral surface is dorsoventrally convex. It forms a long caudoventral process that supports the articular and the ventral surface and participates in the ventral margin of the retroarticular process (Fig. 7). The rostral portion of the ventral border of the angular is straight, then it becomes distinctly concave, so that the caudoventral margin of the mandible looks notched and the mandibule sigmoidal in lateral view. The ventral margin of the angular is slightly concave in *Haya* (*Makovicky et al., 2011*; *Norell & Barta, 2016*), although it is straight to slightly convex in *Hypsilophodon* (*Galton, 1974*), *Jeholosaurus* (*Barrett & Han, 2009*) and *Changchunsaurus* (*Jin et al., 2010*).

**Articular** — The articular is relatively massive and is inserted between the angular ventrally and the surangular laterally. It forms a prominent retroarticular process that extends caudally over 1 cm beyond the caudal margin of the quadrate (Fig. 7).

**Axial skeleton**
The complete vertebral series is preserved and mostly articulated in the holotype specimen. Only the very tip of the tail is potentially missing. However, because the specimen lies on its belly, the vertebrae are still partly embedded within the matrix, and in most cases, only the neural arches and spines are visible. The important extension of ossified tendons

along the dorsals and sacrals and the presence of articulated ribs further complicates the description of the vertebral series.

**Cervical vertebrae and ribs —** The neck of *Changmiania* is extremely shortened, formed by only 6 vertebrae, as observed in both the holotype and referred specimen. In basal ornithischians, ornithopods and ceratopsians, the neck is usually composed of nine vertebrae, as occurs in *Heterodontosaurus* (*Santa Luca, 1980*), *Hexinlusaurus* (*He & Cai, 1984*; *Barrett, Butler & Knoll, 2005*), *Agilisaurus* (*Peng, 1992*), *Hypsilophodon* (*Galton, 1974*), *Jeholosaurus* (*Han et al., 2012*), *Changchunsaurus* (*Butler et al., 2011*), *Haya* (*Makovicky et al., 2011*) and *Orodromeus* (*Scheetz, 1999*). *Psittacosaurus* species possess 8 or 9 cervical vertebrae (*Sereno, 1987*), basal neoceratopsians usually 10 (*You & Dodson, 2003*), and iguanodontians also have more than 9 cervical vertebrae. In ornithischians a reduced number of cervical vertebrae is observed in the thyreophoran lineage, with 8 cervicals in *Scelidosaurus* (*Norman, 2020*), and 7–8 cervicals in ankylosaurs (*Vickarious, Maryańska & Weishampel, 2004*).

Cranially, a discrete atlantal intercentrum is tentatively identified on the left side of the specimen, although it is still nearly completely embedded in the matrix. The centrum of the axis appears longer than that of the succeeding cervical vertebrae (Fig. 8A). Its lateral surface is strongly concave craniocaudally. The diapophysis is visible on the right lateral surface of the centrum, close to the suture with the neural arch. It means that the axial rib (not preserved in this specimen), was double-headed, as in *Agilisaurus* (*Peng, 1992*), *Jeholosaurus* (*Han et al., 2012*), *Haya* (*Makovicky et al., 2011*), *Orodromeus* (*Scheetz, 1999*) and *Psittacosaurus*, contrasting with the single-headed axial ribs in *Hypsilophodon* (*Galton, 1974*) and *Changchunsaurus* (*Butler et al., 2011*). In lateral view, the neural spine is oriented posterodorsally at an angle of about 30 degrees to the horizontal. It is particularly elongate, extending well beyond the caudal border of the axial centrum to overlap cervical vertebra 3, as also observed in *Lesothosaurus* (*Sereno, 1991*), *Heterodontosaurus* (*Santa Luca, 1980*), *Jeholosaurus* (*Han et al., 2012*), *Changchunsaurus* (*Butler et al., 2011*) and *Haya* (*Makovicky et al., 2011*). Along its midline, the spine forms a particularly sharp crest extending along its entire length. The posterior margin of the axial neural spine is expanded transversely to form a frill-like plate above the cranial aspect of cervical 3. The postzygapophyses are set on the laterocaudal corner of the frill and face ventrally.

The longest cervical is cervical 3; then, there is apparently a slight decrease in length until cervical 6, as also noticed in *Changchunsaurus* (*Butler et al., 2011*). The cervical centra appear less elongated than in *Oryctodromeus* (*Krumenacker, 2017*). The lateral surfaces of the centra are concave both anteroposteriorly and dorsoventrally. The neural arch of cervical 3 is much longer than its corresponding centrum, resembling a somewhat reduced version of the axial neural arch (Fig. 8A). Its caudal margin is also slightly expanded transversely, supporting robust ventrally-facing postzygapophyses. The neural arches become progressively shorter through cervicals 3–6. On the contrary, the diapophyses become progressively more robust and elongate. On cervical 4, the diapophyses projects caudoventrally, forming an angle of approximately 45° with
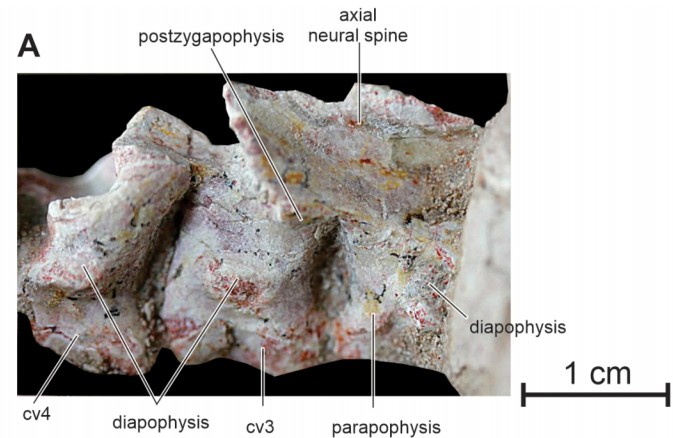

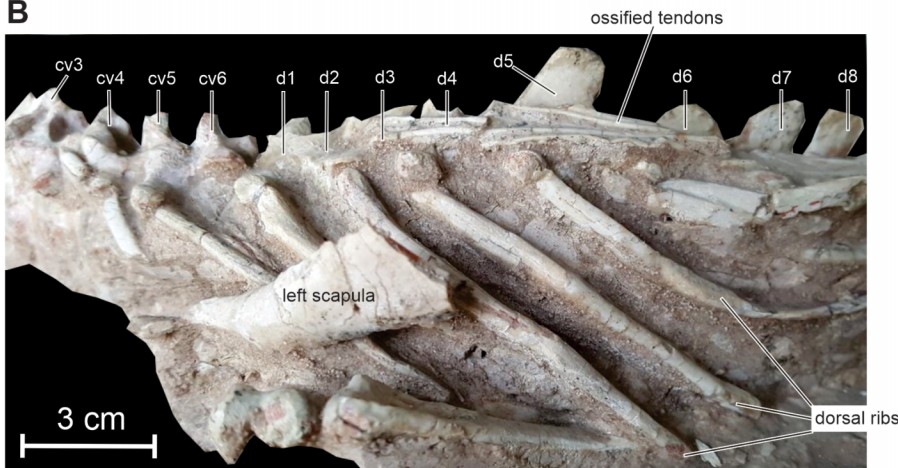

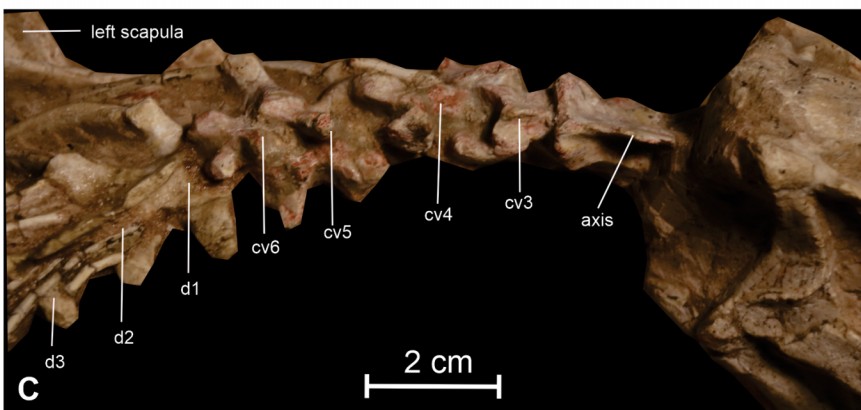

**Figure 8 Axial skeleton of *Changmiania liaoningensis*.** (A) Cervical vertebrae 2 to 4 of PMOL AD00114 in right lateral view; (B) cervical and dorsal series of PMOL AD00114 in left lateral view; (C) neck region of PMOL LFV022 in right dorsolateral view. Abbreviations: cv: cervical vertebra; d: dorsal vertebra.

the horizontal plane in lateral view (Fig. 8A); on cervicals 5 and 6, their orientation gradually changes from caudoventral to caudolateral. At their base, the prezygapophyses are prominent, facing dorsomedially. The neural spine is incipiently developed on cervical

3, as is usual in small ornithischians; it is distinctly better developed both in *Jeholosaurus* (*Han et al., 2012*) and *Changchunsaurus* (*Butler et al., 2011*). The neural spines gradually become more prominent and hook-like on cervicals 4–6 (Figs. 8B and 8C). The postzygapophyses remain large and robust, but progressively face more laterally.

Cervical ribs 4–6 are partially preserved and progressively increase in length. Rib 4 is clearly double-headed (Fig. 8B).

**Dorsal vertebrae and ribs** — All the dorsal vertebrae are concealed by matrix, and their dorsal surface is covered by a dense lattice of ossified tendons (Fig. 8B), so that their description is very limited. There are 15 or 16 dorsal vertebrae in *Changmiania*, as also reported in *Hypsilophodon* (*Galton, 1974*), *Jeholosaurus* (*Han et al., 2012*), *Changchunsaurus* (*Butler et al., 2011*) and *Haya* (*Makovicky et al., 2011*); the dorsal series of heterodontosaurids and basal neoceratopsians appears shorter, with 13 dorsals reported in *Heterodontosaurus* (*Sereno, 2012*) and 12 in *Archaeoceratops* (*You & Dodson, 2003*). The transverse processes lie at the same level as the zygapophyses, as is usual in basal ornithopods (*Norman et al., 2004*). At the cranial end of the dorsal series, the transverse processes are oriented dorsolaterally, but they rapidly shift to a more horizontal orientation further toward the sacrum. The neural spines are rather low; they are inclined caudally at the cranial end of the dorsal series, but progressively become more vertical further toward the sacrum. At the caudal end of the dorsal series, several neural spines appear fused together.

Dorsal ribs are rather robust; the longest lie along the level of dorsals 5–8 (Fig. 8B). The anterior dorsal ribs are dorsoventrally compressed, strongly curved, and are slightly grooved along their cranial and caudal surfaces. The last dorsal rib is notably short and robust and projects laterally; its distal extremity lies about the level of the cranial end of the preacetabular process of the ilium. It is apparently single-headed.

**Sacral vertebrae** — Only the neural spines of the sacral vertebrae are visible on the holotype specimen. They are completely fused together, forming a craniocaudally-elongated continuous bar (Fig. 9A), whose apex lies slightly dorsally to the level of the dorsal edge of the ilium. Such a complete fusion is unusual in small ornithischians, although it might also be explained by ontogeny. The number of sacral vertebrae cannot be adequately estimated in *Changmiania*. Six sacral vertebrae occur in many basal ornithischians, ornithopods and ceratopsians, including *Heterodontosaurus* (*Sereno, 2012*), some specimens of *Hypsilophodon* (*Galton, 1974*), *Jeholosaurus* (*Han et al., 2012*), *Haya* (*Makovicky et al., 2011*), *Orodromeus* (*Scheetz, 1999*), *Parksosaurus*, *Thescelosaurus* (*Norman et al., 2004*), *Psittacosaurus* and *Archaeoceratops* (*You & Dodson, 2003*). Five sacral vertebrae are present in some basal ornithischians, including *Lesothosaurus* (*Sereno, 1991*), *Agilisaurus* (*Peng, 1992*), *Hexinlusaurus* (*He & Cai, 1984*; *Barrett, Butler & Knoll, 2005*), probably *Eocursor* (*Butler, Smith & Norman, 2007*) and some specimens of *Hypsilophodon* (*Galton, 1974*). Sacra with more than six sacral vertebrae occur in *Oryctodromeus*, likely *Orodromeus* (*Varricchio, Martin & Katsura, 2007*),

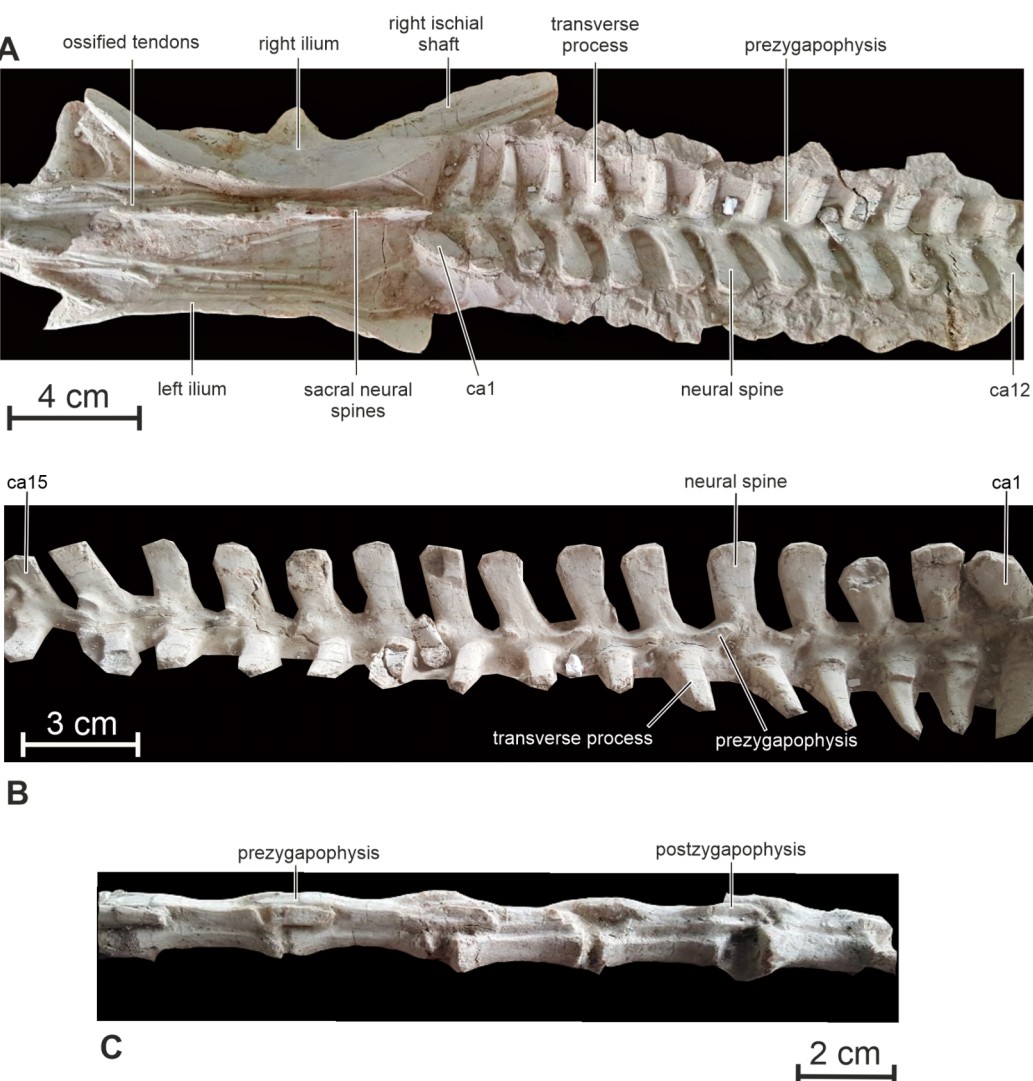

**Figure 9 Axial skeleton of PMOL AD00114.** (A) Sacrum and proximal caudal vertebrae in dorsal view; (B) proximal caudal vertebrae in right lateral view; (C) distal caudal vertebrae in left lateral view. Abbreviation: ca, caudal vertebra.

derived iguanodontians (*Norman, 2004*) and ceratopsians (*Dodson, Forster & Sampson, 2004*).

**Caudal vertebrae** — Thirty-six caudal vertebrae are preserved in the holotype specimen. Only a few distalmost vertebrae are possibly missing. The tail is particularly long, making up to 55% of the total length of the animal, and robust (Figs. 1A and 1C). The neural spines are developed along the 18 proximal vertebrae (Fig. 9B). Their height progressively decreases towards the distal end of the tail. More distally, the neural spine forms a low ridge along the dorsal surface of the neural arch. Overall, the neural spines look more elongated proximodistally than in *Hypsilophodon* (*Galton, 1974*) and *Haya* (*Makovicky et al., 2011*).

The transverse processes of the proximal caudals are particularly prominent, tapering distally to a rounded terminus; they project horizontally and curve slightly posteriorly in their distal part (Fig. 9A). Passing through the caudal series, the transverse processes progressively become shorter, proportionally more massive, with a distal end that is slightly expanded proximodistally. The transverse processes completely disappear at about the level of the 20th caudal. The prezygapophyses extend proximally from the base of the neural arch to cover the postzygapophyses of the preceding vertebra, set at the ventrocaudal corner of the neural spine. The articular facets of the pre-and postzygapophyses are nearly vertical. The centrum of the distal caudal vertebrae becomes progressively more elongated proximodistally, becoming up to 4.5 times as long as high (Fig. 9C). Distal caudals remain proportionally shorter in *Jeholosaurus* (twice as long as high: *Han et al. (2012)*). Both the pre- and postzygapophyses become very elongate, extending well beyond the proximal and distal limits of the centrum and exhibiting considerable overlap. The articular facets of the zygapophyses remain almost vertically inclined.

**Ossified tendons** — The ossified tendons form a particularly dense lattice alongside the neural spines of the dorsal and sacral vertebrae, extending cranially up to the dorsal surface of second dorsal vertebra (Figs. 8B and 8C). They are completely absent from the tail region. The tendons are relatively robust and cylindrical in cross-section. They lack a distinct lattice-like arrangement and appear to be arranged in linear bundles. This distribution of ossified tendons is similar to that in the basal ornithischians *Lesothosaurus* (*Thulborn, 1972*), *Agilisaurus* (*Peng, 1992*) and H*eterodontosaurus* (*Santa Luca, 1980*), some basal ornithopods including *Jeholosaurus* (*Han et al., 2012*) and *Haya* (*Makovicky et al., 2011*). However, there is no evidence for the tendons extending onto anterior dorsal vertebrae in *Jeholosaurus* (*Han et al., 2012*). In *Hypsilophodon* and *Oryctodromeus*, ossified tendons are also present alongside the dorsal and sacral regions, but they extend up to the distal part of the tail (*Galton, 1974*; *Krumenacker, 2017*). In *Psittacosaurus xinjiangensis*, ossified tendons extend on the proximal part of the tail (*Sereno & Zhao, 1988*). In the heterodontosaurid *Tianyulong*, few epaxial ossified tendons are present near the neural arches of the posterior dorsal, sacral and anterior caudal vertebrae, but a sheath of ossified tendons starts around the seventh caudal vertebra, spanning the neural spines and chevrons and considerably stiffening the mid and distal portions of the tail (*Sereno, 2012*).

**Appendicular skeleton**
**Scapula** — Although the scapula and coracoid are completely fused, their respective limits can still be discerned (Figs. 10A–10C). Fused scapulocoracoids can also be observed in *Oryctodromeus* (*Varricchio, Martin & Katsura, 2007*), *Koreanosaurus* (*Huh et al., 2011*), and *Haya* (*Makovicky et al., 2011*). The scapula is bowed medially to follow the contour of the rib cage. Its proximal plate is dorsoventrally expanded. A wide deltoid fossa is developed on the lateral side of the proximal plate of the scapula, being limited by a massive deltoid ridge (Fig. 10C). As in *Zephyrosaurus*, *Orodromeus* (*Scheetz, 1999*),

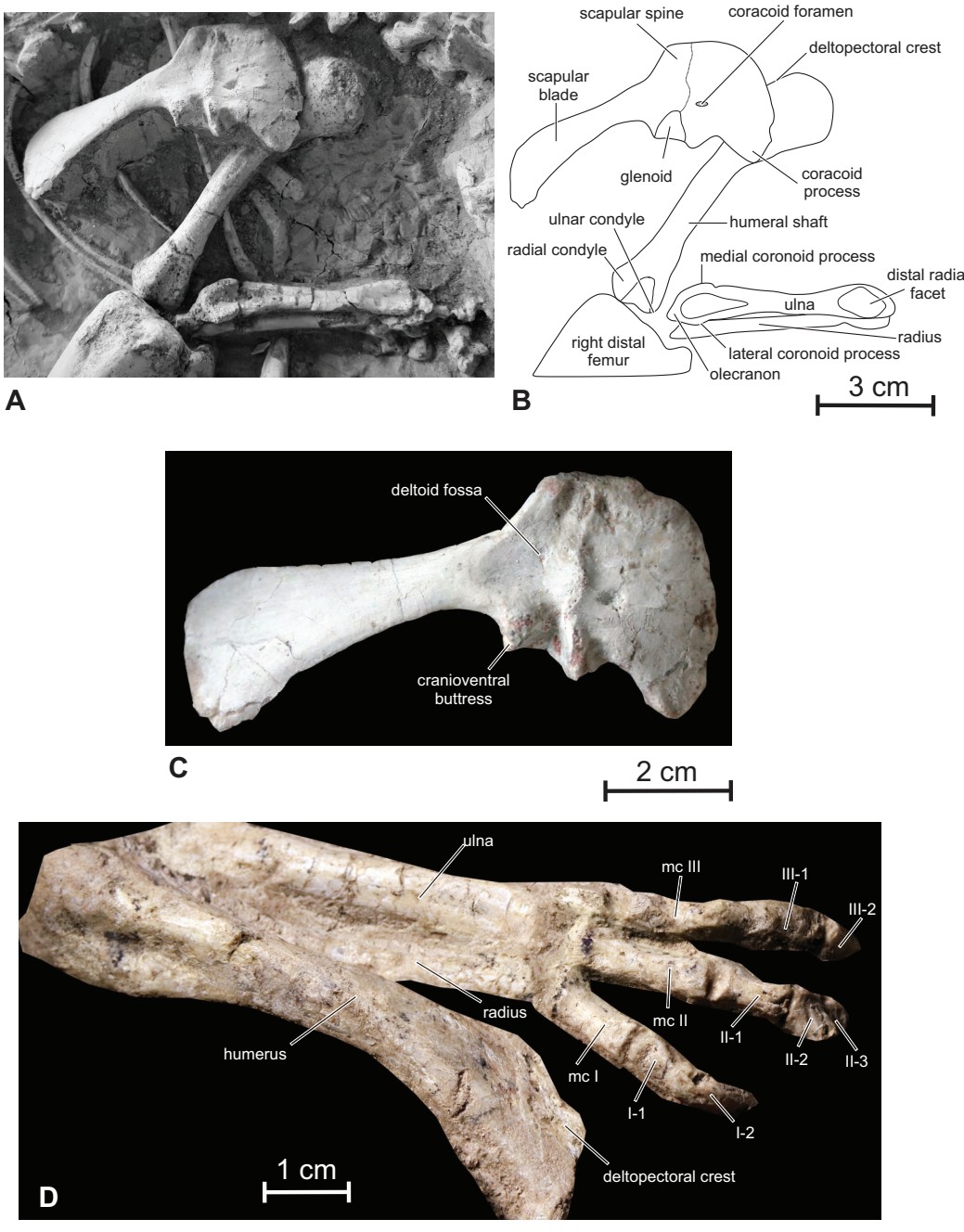

**Figure 10 Scapular girdle and forelimb of *Changmiania liaoningensis*.** (A) Photograph and (B) line drawing of the right scapular girdle and forelimb of PMOL AD00114; (C) detail of the right scapulo-coracoid of PMOL AD00114 in lateral view; (D) left forelimb of PMOL LFV022 in dorsal view. Abbreviation: mc, metacarpal.

*Oryctodromeus* (*Varricchio, Martin & Katsura, 2007*) and *Koreanosaurus* (*Huh et al., 2011*), a strong scapular spine projects craniodorsally. As also observed in *Koreanosau*rus (*Huh et al., 2011*), the cranioventral angle of the scapula forms a large buttress for attachment site of a strong *m. triceps scapularis lateralis externus* With a ratio 'length/ minimal height of the scapula' = 6.2, the scapular blade is also relatively robust. The distal

end of the scapula is better preserved in the referred specimen JLUM LFV022 (Fig. 1C). As is usual in basal ornithopods, the ventral margin of the scapular blade is markedly concave in lateral view; however, the dorsal margin is also concave, although it remains nearly perfectly straight in e.g. *Hypsilophodon* (*Galton, 1974*), *Jeholosaurus* (*Han et al., 2012*), *Changchunsaurus* (*Butler et al., 2011*), *Haya* (*Makovicky et al., 2011*), *Zephyrosaurus*, *Orodromeus* (*Scheetz, 1999*, fig. 19), *Thescelosaurus* (*Gilmore, 1915*) and *Kulindadromeus* (*Godefroit et al., 2014*). Consequently, the distal end of the scapula is expanded both ventrally and dorsally in *Changmiania*, although the ventral expansion remains more important, contrasting with the more asymmetrical ventral expansion in other basal ornithopods. The caudal scapular blade of *Oryctodromeus* is characterized by a much larger surface area relative to the overall size of the scapulocoracoid when compared with other ornithischians and ornithopods, allowing for larger muscles attachment sites (*Fearon & Varricchio, 2016*).

**Coracoid** — The coracoid closely resembles that of *Hypsilophodon* (*Galton, 1974*), *Orodromeus* (*Scheetz, 1999*) and *Changchunsaurus* (*Butler et al., 2011*). In lateral view, it is subquadrate, with a convex cranial margin, and forms a robust cranioventral hook-like process (Figs. 10A–10C). Its ventral margin looks consequently notched. The coracoid forms about half of the glenoid; the glenoid surface of the coracoid is kidney-shaped. The coracoid is transversely expanded caudally at the level of the articular facet for the scapula. The coracoid foramen is small, slit-like, and perforates the bone close to the articular facet for the scapula. The coracoid of *Changmiania* lacks a prominent cranioventrally extending ridge on its lateral surface, developed in *Psittacosaurus* (*Averianov et al., 2006*) and *Yandusaurus* (*He & Cai, 1984*).

**Humerus** — The right humerus is visible in dorsal view and looks rather slender, contrasting with the much stouter humeri in *Oryctodromeus* and *Koreanosaurus*, in which the shafts are relatively shorter and wider relative to the total length of the bone (*Fearon & Varricchio, 2016*; *Huh et al., 2011*). It is distinctly longer than the scapula (Table 1), as in *Kulindadromeus* (*Godefroit et al., 2014*), but unlike in *Jeholosaurus* (P. Godefroit, 2016, personal observation), *Haya* (*Makovicky et al., 2011*), and likely *Oryctodromeus* (*Fearon & Varricchio, 2016*; *Krumenacker, 2017*), in which the scapula and humerus are subequal in length. The proximal portion of the humerus is expanded transversely and compressed craniocaudally (Figs. 10A and 10B), with a regularly rounded proximal border.
The humeral head forms a distinct thickening on the caudal surface, at the midpoint of the proximal border. The deltopectoral crest is obscured by the right coracoid in the holotype. It appears relatively short in the referred specimen (Fig. 10C), about 1/3 the length of the humerus; it is in any case proportionally shorter and less prominent than in *Kulindadromeus* (*Godefroit et al., 2014*). The humeral shaft is slender and slightly compressed craniocaudally, with an elliptical cross-section. The distal end is slightly expanded laterally to form the two articular condyles. The lateral radial condyle and the medial ulnar condyle have similar sizes. The cranial surface of the distal end forms a shallow triangular depressed area (Figs. 10A and 10B).

**Table 1 Selected measurements of *Changmiania liaoningensis* PMOL AD00114 specimen.**

| | |
|---|---|
| Total length: | 1,170 mm |
| Skull, length: | 110.5 mm |
|    Maximum height of skull: | 37.4 mm |
| Orbit length: | 30 mm |
| Snout length: | 52 mm |
| External antorbital fenestra length: | 13 mm |
| External naris length: | 11 mm |
| Supratemporal fenestra length: | 17 mm |
| Infratemporal fenestra, rostrocaudal length of dorsal margin: | 18 mm |
|    Height: | 20 mm |
| Mandible length: | 93 mm |
| Dentary, length: | 52 mm |
|    Height of dentary at mid-length: | 13.5 mm |
| Tail total length: | 650 mm |
| Scapula, length: | 68 mm |
|    Height of proximal head: | 27 mm |
|    Minimum height of blade: | 11 mm |
| Humerus, length: | 85 mm |
|    Minimum mediolateral width: | 8 mm |
| Ulna, length: | 60 mm |
|    Mediolateral width of proximal end: | 11.3 mm |
|    Mesiolateral width of distal end: | 9.3 mm |
| Ilium, length: | 101.5 mm |
|    Length of preacetabular process: | 42 mm |
|    Length of postacetabular process: | 36 mm |
| Femur length: | 115.5 mm |
| Tibotarsus length: | 140 mm |
| Fibula length: | 130 mm |

Individual pes element lengths:

| | | | | |
|---|---|---|---|---|
| Mt I: ? Mt II: ? | Mt III: 67 mm | Mt IV: 60 mm | | |
| I-1: ?  I-2: ? | | | | |
| II-1: 19 mm | II-2: 15.2 mm | II-3: 18.4 mm | | |
| III-1: 21.5 mm | III-2: 15.6 mm | III-3: 15 mm | III-4: > 20 mm | |
| IV-1: 12.5 mm | IV-2: 12 mm | IV-3: 10.6 mm | IV-4: 10 mm | IV-5: 17.8 mm |

**Ulna and radius** — In the holotype, the right ulna is completely preserved and visible in cranial view; the right radius is partly hidden by the ulna (Figs. 10A and 10B). The proximal portion of the left ulna is also visible in cranial view. The ulna and radius are distinctly shorter than the humerus (Table 1) and are rather robust. Both are nearly perfectly straight. The olecranon process of the ulna is moderately developed as is usual in basal ornithopods. On the proximal part of the ulna, the craniomedial coronoid process forms a low and rounded crest that progressively merges with the ulnar shaft.

The craniolateral coronoid process is better developed and triangular in cranial view; its proximal portion overhangs the proximolateral side of the ulna. Between these processes, the articular facet for the proximal part of the radius is large, triangular and deeply concave. The ulna progressively tapers distally. Its distal end is slightly expanded mediolaterally again. Its cranial side forms a smoothly concave facet, slightly inclined laterally for the distal part of the radius.

The radius is visible in the referred specimen PMOL LFV022 (Fig. 10C); it is straight and as robust as the ulna. Its distal end is weakly twisted about its longitudinal axis, as also observed in *Haya* (*Makovicky et al., 2011*).

**Hand** — The right carpus and manus are eroded and partly hidden under the skull in the holotype PMOL AD00114 (Figs. 1A and 1B). The following description is therefore based on the referred specimen PMOL LFV022, which has better preserved hands. Only fingers I to III are visible in dorsal view in this specimen (Fig. 10D). They are proportionally short, about half the length of the humerus, as also observed in *Hypsilophodon* (*Galton, 1974*), contrasting with the proportionally longer and more slender hand in the heterodontosaurids *Heterodontosaurus* (*Santa Luca, 1980*), *Abrictosaurus* and *Tianyulong* (*Sereno, 2012*). Metacarpals I–III are short and robust; as in *Hypsilophodon* (*Galton, 1974*), metacarpal III is the smallest, followed by metacarpals II and I. In *Heterodontosaurus* and *Tianyulong*, metacarpal II is the longest, followed by metacarpals III and I (*Sereno, 2012*). The distal ends of metacarpal I–III appear slightly divergent, although they appear more closely packed, running parallel with each other, in *Hypsilophodon* (*Galton, 1974*). The phalangeal count is 2-3-?-?-?, as usual in basal ornithischians including *Abrictosaurus*, *Hypsilophodon* and *Hexinlusaurus* (*Norman et al., 2004*). Phalanges II-2 and III-2 are particularly short. The ungual of digit I appears rather massive and recurved towards the palmar side of the hand.

**Ilium** — The ilium of *Changmiania* closely resembles that of the basal neoceratopsian *Auroraceratops* (*Morschhauser et al., 2019*, fig. 21). Its dorsal margin is regularly convex along the whole length of the bone (Fig. 11A). It is straighter above the main plate and postacetabular process e.g. in *Kulindadromeus* (*Godefroit et al., 2014*), *Haya* (*Makovicky et al., 2011*), *Jeholosaurus* (*Han et al., 2012*) and *Thescelosaurus* (*Gilmore, 1915*). As also described in *Kulindadromeus* (*Godefroit et al., 2014*), the preacetabular process is about 40% of the ilium length, dorsoventrally narrow and strongly deflected ventrally, reaching the level of the pubic peduncle. The preacetabular process is distinctly shorter and more slender in *Oryctodromeus* (*Krumenacker, 2017*). As in *Jeholosaurus* (*Han et al., 2012*), *Haya* (*Makovicky et al., 2011*), *Hexinlusaurus* (*He & Cai, 1984*) and *Hypsilophodon* (*Galton, 1974*), the postacetabular process is slightly shorter, but dorsoventrally much higher than the preacetabular process, contrasting with the dorsoventrally narrow postacetabular process in *Kulindadromeus* (*Godefroit et al., 2014*). In *Orodromeus*, the postacetabular process is relatively longer than the preacetabular process (*Scheetz, 1999*), whereas the postacetabular process of *Heterodontosaurus* accounts for only ~25% of total ilium length (*Santa Luca, 1980*). There is no trace of a

supraacetabular crest along its lateral surface. As in *Kulindadromeus* (*Godefroit et al., 2014*), *Haya* (*Makovicky et al., 2011*) and *Auroraceratops* (*Morschhauser et al., 2019*), the pubic peduncle is prominent, triangular in lateral view, and forms a 45° angle with the craniocaudal axis of the ilium body; the pubic peduncle is usually much reduced in most Cerapoda (*Ösi et al., 2012*). The ischiac peduncle projects ventrally and is stouter than the pubic peduncle. The acetabulum is deep and semi-circular, without any trace of a supraacetabular flange and of a medioventral acetabular flange. As in *Kulindadromeus* (*Godefroit et al., 2014*), *Jeholosaurus* (*Han et al., 2012*), *Changchunsaurus* (*Butler et al., 2011*), *Hypsilophodon* (*Galton, 1974*) and *Orodromeus* (*Scheetz, 1999*), the posterior portion of the brevis shelf cannot be observed in lateral view; it contrasts with the condition in basal ornithischians such as *Agilisaurus* (*Peng, 1992*), *Scelidosaurus*, *Lesothosaurus* (*Butler, 2005*) and *Haya* (*Makovicky et al., 2011*), in which the brevis fossa is visible in lateral view. In both the holotype and referred specimen of *Changmiania liaoningensis*, the paired ilia are not perfectly vertical, as it is the case in most ornithischians, but they are symmetrically inclined dorsomedially, partially covering the sacrum in dorsal view. This inclination is probably not a taphonomic artifact, because the ilia are symmetrically inclined on both the holotype and referred specimen.

**Ischium** — Only the distal part of the paired ischial shafts is visible. They are dorsoventrally flattened and slightly expanded transversely, forming a distal ischial symphysis (Figs. 11B and 11C). Unfortunately, the precise extent of the distal symphysis cannot be accurately estimated in *Changmiania*. Elongate ischiac symphyses occur in *Lesothosaurus* (*Butler, 2005*), *Eocursor* (*Butler, 2010*), *Jeholosaurus* (*Han et al., 2012*), and likely in *Haya* (*Makovicky et al., 2011*). In *Agilisaurus*, the symphysis accounts for approximately 50% of ischial length (*Barrett, Butler & Knoll, 2005*), although the symphysis is much shorter in *Hypsilophodon* (*Galton, 1974*).

The pubis is covered by sediment and thus not visible in both the holotype and referred specimen of *Changmiania liaoningensis*.

**Femur** — Both femora are visible in cranial, medial, and lateral view in both the holotype and referred specimen of *Changmiania liaoningensis*. Their proximal end is hidden under the ilia (Figs. 1A–1C). The femur of *Changmiania* is rather robust and slightly bowed cranially (Figs. 11B and 11C). On the proximal portion of the femur, the lesser trochanter is eroded, but appears moderately developed and supported by a low ridge on the laterocranial corner of the proximal part of the femur; its apex lies well below the apex of the greater trochanter and the intertrochanteric cleft seems poorly developed. The cranial aspect of the femoral shaft is very convex. The fourth trochanter is covered by sediments in both the holotype and referred specimen of *Changmiania liaoningensis*. The distal end of the femur is slightly expanded laterally and compressed craniocaudally. The medial condyle is stouter than the lateral condyle. There is a shallowly concave triangular surface on the cranial side of the distal femur, but there is no real extensor intercondylar groove (Figs. 10B and 10C); this condition is also encountered in the basal ornithopods *Hypsilophodon* (*Galton, 1974*: fig. 54), *Orodromeus* (*Scheetz, 1999*),

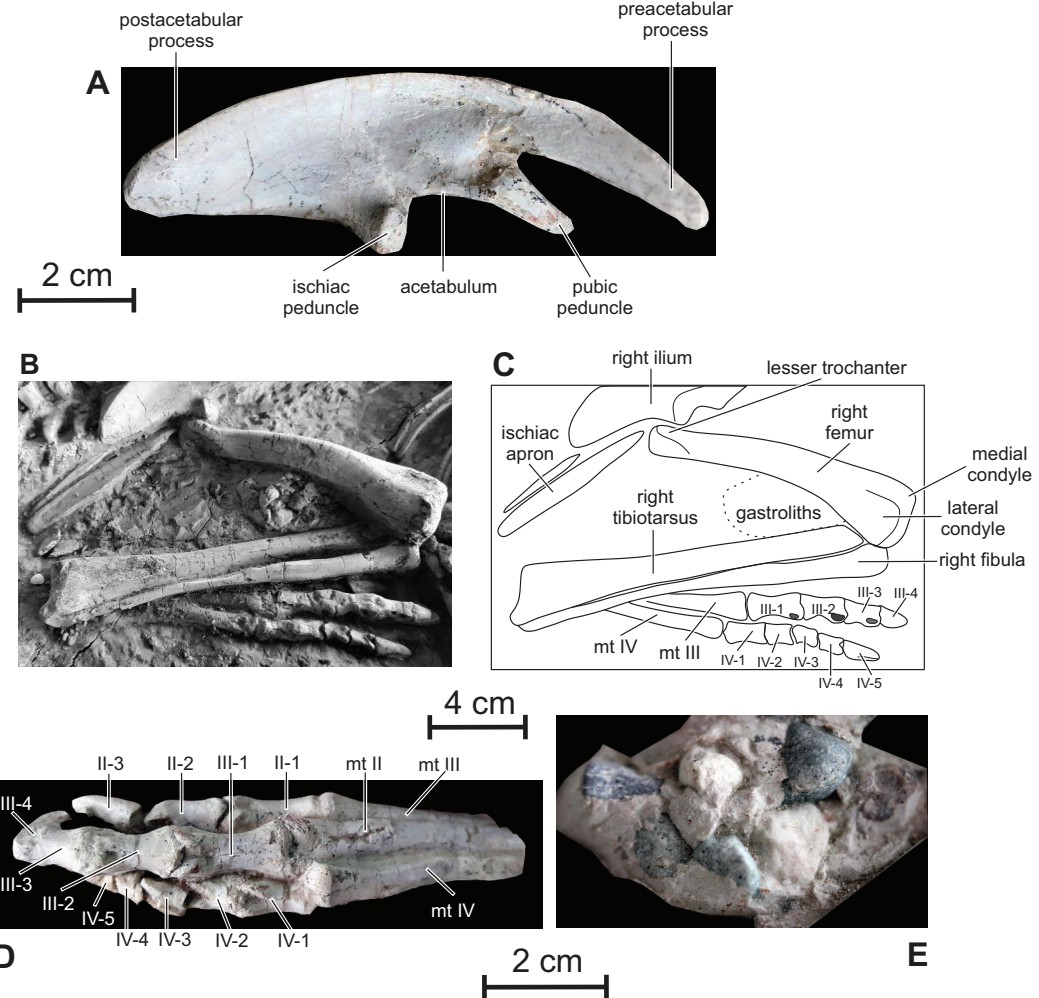

**Figure 11 Pelvic girdle and hindlimbs of PMOL AD00114.** (A) Right ilium in lateral view; (B) photograph of distal ischia and right hindlimb; (C) line drawing of distal ischia and right hindlimb; (D) left foot in dorsal view; (E) gastroliths.

*Jeholosaurus* (*Han et al., 2012*), *Changchunsaurus* (*Butler et al., 2011*) and *Koreanosaurus* (*Huh et al., 2011*). This groove is much better developed in more derived ornithopods (*Butler, Upchurch & Norman, 2008*).

**Tibiotarsus** — Both tibiae are visible in dorsal view in both the holotype and referred specimen of *Changmiania liaoningensis* (Figs. 1A–1C); their proximal end is hidden by the overlapping femora. The astragalus and calcaneum are completely fused to the distal end of the tibia so their respective limits can be hardly discerned. The tibia and tarsal element remain free elements in other basal ornithopods, including *Hypsilophodon* (*Galton, 1974*), Orodromeus (*Scheetz, 1999*), *Haya* (*Makovicky et al., 2011*), *Changchunsaurus* (*Butler et al., 2011*), *Jeholosaurus* (*Han et al., 2012*) and *Oryctodromeus* (*Krumenacker, 2017*). However, fusion of the tarsal elements to the tibia is regarded as an ontogenetical character in hadrosaurid ornithopods (*Godefroit, Bolotsky & Bolotsky, 2012*); thus, pending further evidence, we have not included the fusion of the tibiotarsus

**Table 2 Comparisons of postcranial measurements (in mm) in selected basal ornithopods: *Changmiania liaoningensis, Dryosaurus altus, Koreanosaurus boseongensis, Haya griva, Hypsilophodon foxii, Jeholosaurus shangyuanensis, Nanosaurus agilis, Orodromeus makelai* and *Oryctodromeus cubicularis*.**

| Taxon | Ref | Humerus length | Femur length | Tibia length | H/F | T/F |
|---|---|---|---|---|---|---|
| *C. liaoningensis* | JMOL AD00114 | 85 | 115.5 | 140 | 0.74 | 1.21 |
| *K. boseongensis* | KDRC-BB | 205–215 | 196.5 | 204 | >1 | 1.04 |
| *H. foxii* | NHMUK R5829 | 159 | 198 | 238 | 0.8 | 1.20 |
| *H. foxii* | NHMUK R5830 | 74 | 101 | 117 | 0.74 | 1.16 |
| *O. makelai* | MOR 294 | 72 | 106 | 132 | 0.68 | 1.25 |
| *O. cubicularis* | MOR 1636 | 157 | – | 254 | – | – |
| *J. shangyuanensis* | IVPP V12542 | 61.5 | 94.4 | 113.4 | 0.65 | 1.20 |
| *H. griva* | IGM 100/2015 | 86 | 131 | 155 | 0.66 | 1.18 |
| *N. agilis* | BYU ESM-163R | 104 | 151 | 180 | 0.69 | 1.19 |
| *D. altus* | YPM 1876 | 190 | 360 | 395 | 0.53 | 1.1 |

elements as a diagnostic character for *Changmiania*. The tibiotarsus of *Changmiania* is slender and about 120% the length of the femur as in most basal ornithopods except *Koreanosaurus* (see Table 2). The caudal surface of the slender tibial shaft is regularly convex. Its distal end is flattened craniocaudally and slightly widened mediolaterally, with a shallowly concave caudal surface. The distal end of the tibiotarsus is regularly convex craniocaudally and concave mediolaterally. It is more expanded distally at the level of the calcaneum and external malleolus of the tibia than at the level of the astragalus and internal malleolus of the tibia (Figs. 11B and 11C). The medial surface of the astragalus is flattened and inclined caudolaterally. The lateral surface of the calcaneum is depressed; its dorsal border is concave in lateral view, to fit the rounded distal end of the fibula.

**Fibula** — The proximal half of the fibula is as robust as the corresponding portion of the tibia (Figs. 11B and 11C), contrasting with the slender, rod-like, and distally-tapering fibula observed in most other basal ornithopods, including *Hypsilophodon* (Galton, 1974), Orodromeus (Scheetz, 1999), *Haya* (Makovicky et al., 2011), *Changchunsaurus* (Butler et al., 2011), *Jeholosaurus* (Han et al., 2012) and *Oryctodromeus* (Krumenacker, 2017). Its proximal end is craniocaudally expanded and transversely compressed. It regularly tapers distally, but its distal end is slightly expanded transversely again, with a rounded distal surface, to fit against the dorsal border of the calcaneum. The lateral side of the fibula is regularly convex craniocaudally, whereas its medial surface, which fits against the tibia, is deeply concave.

**Pes** — Both pedes are visible in dorsal view (Figs. 11B–11D). They are likely complete, but their medial part is covered by the tibiotarsus and fibula. Left digit I is partly exposed on the right foot of the referred specimen. It is particularly short, extending up to the distal border of metatarsal II, and formed by two phalanges. In neither pes is it possible to ascertain the presence or absence of metatarsal 5. The pedal count is thus: 2-3-4-5-?.

The metatarsals are tightly appressed to one another (Fig. 11D). Metatarsals II and IV are nearly equal in length and slightly shorter than metatarsal III. Metatarsal III and, especially, metatarsal IV curve laterally in cranial view. The distal articular condyles of the main metatarsals are shallowly developed on metatarsals II and IV, but metatarsal III has distinct lateral and medial condyles. The depression for attachment of the collateral ligaments is shallow on the medial surface of metatarsal II, whereas it is much more pronounced on both the medial and lateral sides of metatarsal III, and on the lateral side of metatarsal IV.

The phalanges of pes digits 2–4 have well-developed collateral ligament pits, dorsal extensor pits, and intercondylar processes (Figs. 11B–11D). Their enlarged proximal articular surfaces are subdivided by a median ridge into two concavities. The ungual phalanges of digits 2–4 are elongate, exceeding the other phalanges in length, with deep lateral ligament grooves and sharp distal ends. The ungual of digit 3 is the longest (Fig. 11D), unlike in *Changchunsaurus*, in which the ungual of digit 2 slightly exceeds the ungual of digit 3 in length (*Butler et al., 2011*).The ungual of digit 1 is too incompletely preserved in the referred specimen to be adequately described.

**Gastroliths** — A clustered mass of a dozen small pebbles is visible between the distal part of the right femur and the proximal part of the tibia in PMOL AD00114 (Fig. 1B). Individual clasts, which range in size from 5 mm to 13 mm, appear to be quartz or chert and are subangular in shape with a smooth patina (Fig. 11E). They closely resemble the pebble clusters associated with the holotype specimen of *Haya griva* (*Makovicky et al., 2011*).

## PHYLOGENETIC ANALYSIS

Our phylogenetic analysis recovered 48 most parsimonious trees, with a length of 859 steps, a Consistency Index (CI) of 0.36 and a Retention Index (RI) of 0.62. The strict consensus of these trees is already well resolved, although the Bremer support remains low for most of the nodes (Fig. 12). This low support is mainly caused by various homoplasies (some of which are functionally significant), which are distributed widely across ornithischian phylogeny. A fully resolved agreement subtree was obtained after the exclusion of 5 unstable 'wilcard' taxa (carried out using the 'Comparisons-Agreement subtree(s)' option of TNT; Fig. 12).

*Changmiania* is placed at the base of the clade Ornithopoda, characterized by the following unambiguous synapomorphies (Tables S2 and S4): rugosities are present on the rostral and dorsal surfaces of the premaxilla; there is a fossa-like depression on the premaxilla–maxilla boundary; the frontals are mediolaterally narrow and rostrocaudally elongate, at least twice as long as wide; in lateral view, the jugal extends ventrally above the distal condyles of the quadrate; and the olecranon process of the ulna is moderately developed.

However, *Changmiania* lacks the following synapomorphies, characteristic for more derived ornithopods: the frontals are dorsally flattened over the orbit (they are still slightly arched in *Changmiania*); the rostral tip of the dentary is positioned at mid-height in lateral

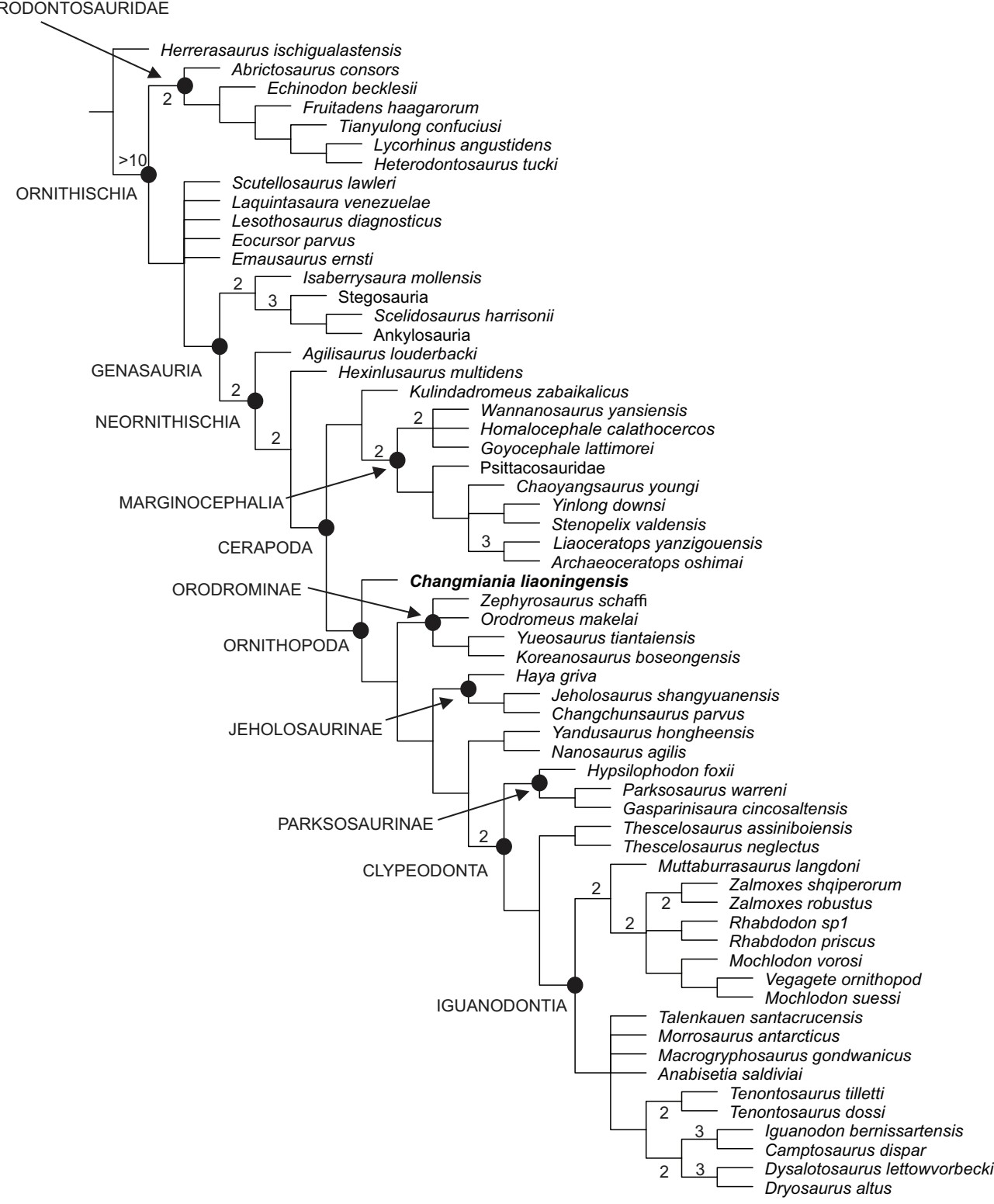

**Figure 12 Phylogenetic position of *Changmiania liaoningensis* gen. et sp. nov. among Ornithischia.** Strict consensus tree of 49 MPT's. Tree Length = 859. Nodal support (Bremer indices >1) is indicated above or below the branches.

view (positioned above mid-height in *Changmiania*); the caudal neural spines extend beyond their own centrum (entirely positioned over the centrum in *Changmiania*); the scapula is longer or subequal to the humerus (the humerus is substantially longer than the scapula in *Changmiania*); and the ischiac peduncle of the ilium is broadly swollen and projects ventrolaterally (it is much reduced and projects ventrally in *Changmiania*).

*Jeholosaurus shangyuanensis*, also from the Lujiatun Beds of western Liaoning, is placed in the clade Jeholosaurinae together with *Haya griva* and *Changchunsaurus parvus* (Figs. 12 and 13). Jeholosaurinae is characterized by the following synapomorphies (Tables S2 and S4): a forked caudal ramus of the jugal (dorsocaudal projection absent in *Changmiania*) and relatively wide coracoid (L/W < 0.6; > 0.7 in *Changmiania*).

Jeholosaurinae share with more derived ornithopods the following synapomorphies (Tables S2 and S4): rod-like supraorbitals (in *Changmiania*, the supraorbitals have a much wider base); the presence of a quadratojugal foramen (absent in *Changmiania*); the longest caudal process is distal to the first caudal vertebra (at the level of the first caudal in *Changmiania*); and the proximal and distal edges of the cranioventral buttress form an acute, less than 75° angle (>75° in *Changmiania*).

The present analysis recovers *Kulindadromeus zabaikalicus*, from the Middle Jurassic of Siberia (*Godefroit et al., 2014*), as the sister-taxon of Marginocephalia (Figs. 12 and 13), with whom it shares the following unambiguous synapomorphies: the humeral shaft is strongly bowed laterally along its length in cranial and caudal views (it relatively straighter in *Changmiania*), and the pubic and iliac peduncles of the ischium are subequal in size, or the iliac peduncle is larger than the pubic peduncle (polarity unknown in *Changmiania*). Using the same dataset of the main analysis, we tested the alternative placement of *Kulindadromeus* as the sister-taxon of Cerapoda, as originally hypothesized by *Godefroit et al. (2014)*; this analysis produced shortest trees that are two steps longer than the shortest trees resulted by the unconstrained analysis.

The results of the present phylogenetic analysis are not quite different from most of other recently proposed analyses, including *Buchholz (2002)*, *Butler, Upchurch & Norman (2008)*, *Butler et al. (2011)*, *Makovicky et al. (2011)*, *Han et al. (2012)*, *Godefroit et al. (2014)* and *Dieudonné et al. (2016)*: all those analyses recover Orodrominae, Jeholosaurinae, *Parksosaurus* and *Thescelosaurus* within the clade Ornithopoda. On the contrary, *Boyd (2015)* places Orodrominae, Jeholosaurinae, *Parksosaurus* and *Thescelosaurus* outside Cerapoda, so that the content of Ornithopoda *sensu Boyd (2015)* is dramatically reduced, limited to *Hypsilophodon* and Iguanodondia (except Elasmaria, also recovered outside Cerapoda). It must be noted that the definitions of these clades proposed by *Boyd (2015)* is nearly identical as those proposed by the other recent analyses, so the different results can only be interpreted by differences in the character-taxon matrix. As already indicated in the Material and Methods section of this paper, we have used an extensively modified version of the character-taxon matrix published by *Dieudonné et al. (2016)*, which had already incorporated *Boyd's (2015)* data. Cerapodan relationships will be discussed in more details in a forthcoming paper (*Dieudonné et al., 2020*).

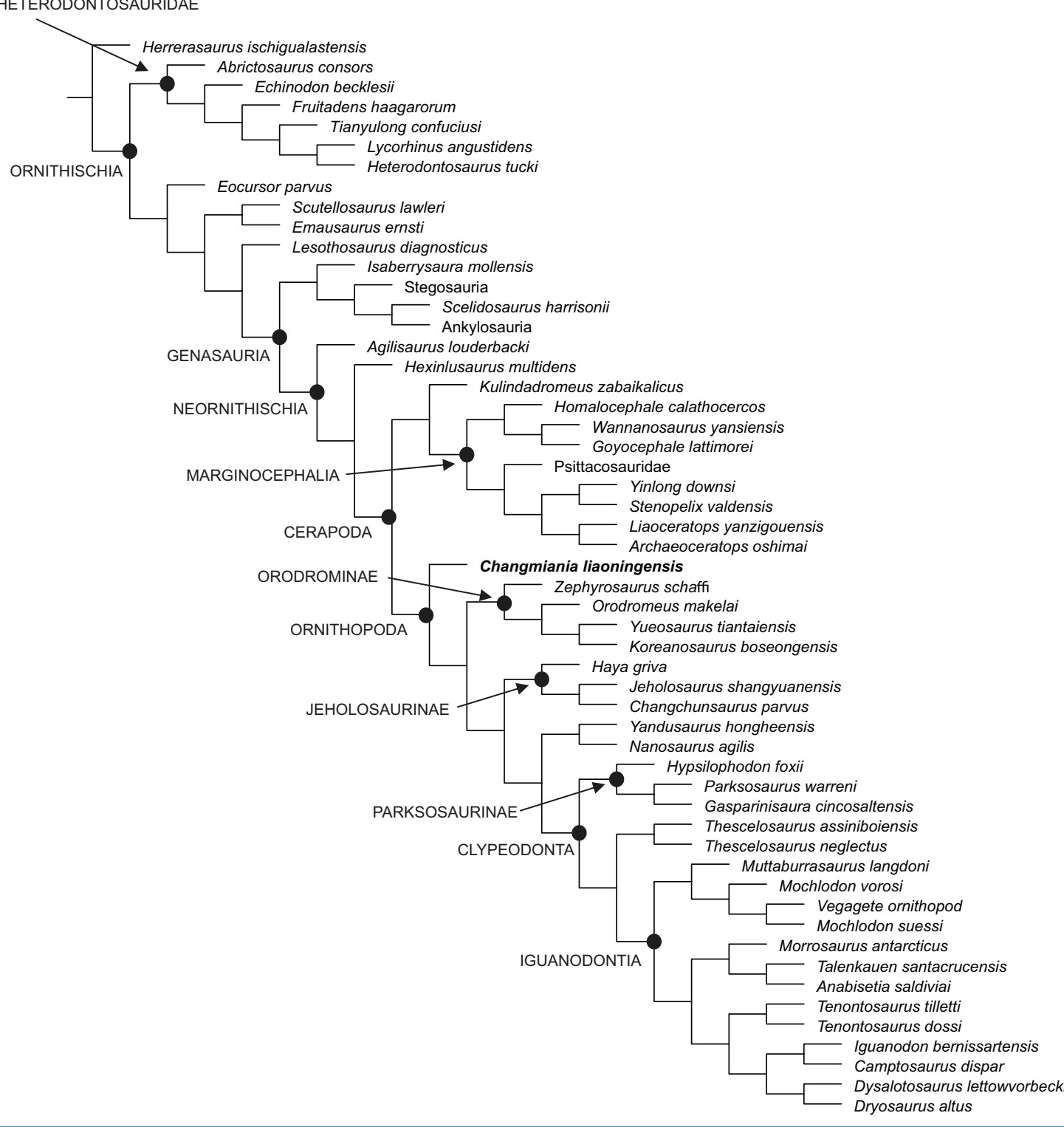

**Figure 13 Phylogenetic position of *Changmiania liaoningensis* gen. et sp. nov. among Ornithischia.** Fully-resolved agreement subtree obtained after the exclusion of five unstable 'wilcard' taxa, carried out using the 'Comparisons-Agreement subtree(s)' option of TN. Fully-resolved agreement subtree obtained after the exclusion of five unstable 'wilcard' taxa, carried out using the 'Comparisons-Agreement subtree(s)' option of TNT (*Goloboff, Farris & Nixon, 2008*).

## DISCUSSION

Vertebrate fossils unearthed from the lower Lujiatun Beds often retain a three-dimensional form, remaining perfectly articulated without any trace of weathering, scavenging or other disturbance (*Meng et al., 2004*; *Zhao, Barrett & Eberth, 2007*). Moreover, some spectacular specimens from the Lujiatun Beds record exceptional behavioural information, such as evidence of parental care in the basal ceratopsian *Psittacosaurus* (*Meng et al., 2004*; *Zhao, Barrett & Eberth, 2007*) and avian-like sleeping posture in the troodontid *Mei* (*Xu & Norell, 2004*; *Gao et al., 2012*). 'Sleeping' *Mei* specimens display a stereotypical sleeping or resting posture found in living birds: the tail curls forward and under the neck, both the hind and fore limbs lie folded beneath the body, and the neck and head curve back between the shoulder and folded elbow toward the hindlimb (*Xu & Norell, 2004*; *Gao et al., 2012*). Although the three-dimensional position of both the *Changmiania liaoningensis* holotype and referred specimens is markedly different from the 'sleeping' *Mei long* specimens and from the stereotypical resting posture in living birds, they share some characteristics also suggesting a resting posture: their body sits on their symmetrically folded hindlimbs, their forelimbs are also symmetrically folded next to their body, with their elbows slightly displaced laterally relative to the trunk (*Xu & Norell, 2004*; *Gao et al., 2012*). Moreover, the trunk and the head of PMOL AD00114 curve laterocaudally, so that its head rests on its right manus. The neck of *Mei long* was much longer and flexible, so that the head of the 'sleeping' specimens curves back between the shoulder and folded elbow toward the hindlimb (*Xu & Norell, 2004*; *Gao et al., 2012*). The main difference between the *Changmiania* and *Mei* 'sleeping' specimens is the orientation of their tail: in *Mei*, it curls forward and under the neck, whereas it remains nearly perfectly straight, oriented backward in both *Changmiania* individuals. It must be noted that the tail of *Changmiania* was likely a rather rigid structure, with limited lateral flexibility: the transverse processes of the proximal caudals are particularly elongated, massive and curved backward, while the pre-and postzygapophyses of the distal caudals are particularly elongate, extending well beyond the proximal and distal limits of the centrum and exhibiting considerable overlap. Curling its tail under its neck in a *Mei*-like style was therefore likely impossible for *Changmiania*.

Such a perfect preservation of the skeleton in a lifelike posture, as observed in both the holotype and referred specimen of *Changmiania liaoningensis* and also in countless fossils from the Lujiatun Beds, implies that the animals were rapidly entombed while they were still alive (*Meng et al., 2004*; *Zhao, Barrett & Eberth, 2007*). Direct sedimentological investigation of the available *Changmiania* specimens is unfortunately impossible because they are already too heavily prepared, and we lack information about their discovery context. *Zhao, Barrett & Eberth (2007)* showed that the sediments entombing a herd of juvenile *Psittacosaurus* skeletons from the Lujiatun Beds near Liu Tai village in western Liaoning represent a lahar (volcanic mudflow), either during the eruptive phase of a nearby volcanic center, or during a non-eruptive debris flow that reworked previously deposited volcanic material. This scenario might be extended to other exceptionally preserved specimens from the same beds, but again we lack precise

stratigraphic, sedimentological and taphonomic information about those specimens as most were discovered by local farmers and most of them were heavily restored. *Gao et al. (2012)* argued that lifelike posture of vertebrate fossils from the Lujiatun beds contradicts burial in a low energy lahar at least for those specimens: even a low energy lahar is likely to move a small animal out of a perimortem position.

It has also been proposed that some of the most fossiliferous locations in the Yixian Formation, and particularly in the Lujiatun Beds, are the result of instant catastrophic mass mortality events preserved in tuffaceous ashes (*Chang et al., 2008*; *Gao et al., 2012*; *Xu & Norell, 2004*); in such a Pompeii-like scenario, the main cause of death for the vertebrates from the Lujiatun beds would have been asphyxiation by toxic volcanic gases of ashes. This hypothesis implies a brief but painful agony, which is certainly not supported by the apparent peaceful posture of the *Mei* and *Changmiania* specimens. *Faux & Padian (2007)* suggest that death resulting from asphyxiation, toxins and infections can be reflected by an opisthotonic posture (hyperextension of the spine with both neck and tail recurved over the back) of the fossil skeleton; this posture is frequent in the two-dimensionally preserved specimens from the Jianshangou and Dawangzhangzi beds of the Yixian Formation, but is markedly different from the 'sleeping' dinosaurs from the Lujiatun Beds. However, *Reisdorf & Wuttke (2012)* interpret the opithotonic posture as a post-mortem rather than a perimortem phenomenom, depending on the varying decay resistance of the soft tissues. *Gao et al. (2012)* similarly show that the troodontid specimens (and also the *Changmiania* fossils) from the Lujiatun Beds lack the 'pugulistic' hyperflexion of hands and toes often observed in humans that perished in pyroclastic flows and surges at Pompeii and Herculanum, interpreted as a perimortem response to death from high temperatures and fire (*Mastolorenzo et al., 2010*).

*Meng et al. (2004)* and *Gao et al. (2012)* propose that sudden entrapment in a collapsed underground burrow might be an alternative mechanism explaining the preservation of lifelike postures in small dinosaurs together with the complete absence of weathering and scavenging traces. *Varricchio, Martin & Katsura (2007)* described fossils of an adult and two juvenile individuals of the basal ornithopod *Oryctodromeus*, discovered in the expanded distal chamber of a sediment-filled burrow. Based on morphological modifications of its skull and postcranial skeleton, *Varricchio, Martin & Katsura (2007)* convincingly hypothesized that *Oryctodromeus* was a burrowing dinosaur. Additional burrows containing *Oryctodromeus* skeletons were subsequently described in Idaho and Montana, reinforcing previous suggestions of communal and fossorial lifestyle, and even possibly parental care in *Oryctodromeus* (*Krumenacker et al., 2019*). Skeletal features observed in *Zephyrosaurus*, *Orodromeus* and *Koreanosaurus*, together with sedimentological and taphonomic data, suggest that these basal ornithopods might also been specialized for digging (*Varricchio, Martin & Katsura, 2007*; *Huh et al., 2011*). The phylogenetic analysis presented here (Figs. 12 and 13) places *Zephyrosaurus*, *Orodromeus* and *Koreanosaurus* in a monophyletic clade named Orodrominae. As already pointed out, *Oryctodromeus* has not been included in the present phylogenetic analysis. However, previous analyses strongly suggest that *Oryctodromeus* is also an orodromine (*Varricchio, Martin & Katsura, 2007*; *Boyd, 2015*; *Krumenacker, 2017*). *Brown et al. (2013)*

also place *Albertadromeus syntarsus*, from the Oldman Fm of Alberta (Canada) amongst Orodrominae. However, this taxon is too fragmentary to be included in the present analysis.

*Changmiania* also displays a series of osteological features that are compatible with a fossorial behaviour in this basal ornithopod. Some extant fossorial vertebrates (e.g. the mole-rat *Spalax*) dig with their head to some degree, using the top of their broad, firm heads to move, loosen, or compact soil (*Hildebrand, 1985*). The fused premaxillae and the spatulate shape of the dorsal surface of the snout in *Changmiania* could represent such an implement. Among ornithopods, fused premaxillae are also present in the orodromine *Zephyrosaurus* and *Oryctodromeus* (*Sues, 1980*; *Varricchio, Martin & Katsura, 2007*; *Krumenacker, 2017*), and in *Changchunsaurus* (*Jin et al., 2010*). However, nuchal crests are particularly developed in head-lifting diggers (*Hildebrand, 1985*), which is not the case in *Changmiania*.

The postcranial skeleton of *Changmiania* shares a series of morphological characteristics with actual scratch-digging mammals, including a shortened neck (six cervical vertebrae), a radius that is significantly shorter (70%) than the humerus, and short hands (*Hildebrand, 1985*; *Elissamburu & De Santis, 2011*). Unfortunately, the ventral side of the ulna is hidden by sediments in both the holotype and referred specimens of *Changmiania*: in *Koreanosaurus* and *Orodromeus*, the proximal ulna is highly keeled as in Talpidae and in the anteater *Tamandua*, providing large insertion areas to both extensor and flexor muscles for the carpus and the digits, in relation with the digging function of the hands (*Lessertisseur & Saban, 1967*; *Hildebrand, 1985*; *Castiella et al., 1992*; *Huh et al., 2011*). The enlarged, fused scapulocoracoid with prominent acromion and scapular spine, present in *Changmiania*, *Koreanosaurus* and *Oryctodromeus* would have increased attachment areas for muscles that were used to stabilize and operate digging forelimbs (*Hildebrand, 1985*; *Varricchio, Martin & Katsura, 2007*; *Huh et al., 2011*; *Fearon & Varricchio, 2016*).

The hip of *Changmiania* exhibits some features that might also tentatively be related to a digging behaviour. Actual mammals that dig with the forefeet usually brace with their hindfeet, often supplemented by the tail serving as a prop (*Hildebrand, 1985*). Burrowing mammals such as the marsupial mole *Notoryctes* and, especially, Talpidae are able to keep their thighs relatively outspread from the axis of their body for a firmer bracing of the back of their body while digging; the pelvis of Talpidae is roughly horizontally oriented, nearly parallel with the vertebral column, and their acetabulum is consequently positioned higher, preventing torsion while bracing (*Lessertisseur & Saban, 1967*). In a similar way, the dorsomedial inclination of its paired ilia above its sacrum potentially helped *Changmiania* to keep its hindlimbs outspread and prevented torsion when digging.

The neural spines of the sacral vertebrae are completely fused together in *Changmiania*, forming a craniocaudally-elongated continuous bar. Reinforcement of the sacropelvic complex, as observed in *Changmiania*, is also a distinctive feature of scratch-digging mammals and appears to relate to the forces, well in excess of body weight, that converge on the pelvic girdle during bracing with the hind feet (*Hildebrand, 1985*). The pelvis of

*Zephyrosaurus*, *Oryctodromeus* and *Orodromeus* is further reinforced by a direct pubosacral contact, but this region is hidden by sediment in both available *Changmiania* specimens. As is frequently observed in burrowers (*Hildebrand, 1985*; *Varricchio, Martin & Katsura, 2007*), the basal portion of the tail is particularly robust in *Changmiania*, with elongated transverse processes and a high neural spine on the proximal caudal vertebrae.

The hindlimb of *Changmiania* is about twice as long as its forelimb and its tibia is significantly longer than its femur, as in most other small basal ornithopods except *Koreanosaurus* (Table 2); those hindlimb proportions suggest that *Changmiania* basically remained an efficient cursorial dinosaur. Moreover, the forelimb and skull modifications remain rather modest, so that *Changmiania* was obviously not a true subterranean animal, but more likely a facultative digger, as also suggested for *Oryctodromeus* (*Varricchio, Martin & Katsura, 2007*; *Krumenacker, 2017*; *Fearon & Varricchio, 2016*; *Krumenacker et al., 2019*).

A possible fossorial behavior in *Changmiania* therefore supports the hypothesis that both PMOL AD00114 and PMOL LFV022 were suddenly entrapped in a collapsed underground burrow, which would explain their perfect lifelike postures and the complete absence of weathering and scavenging traces, as proposed by *Meng et al. (2004)* and *Gao et al. (2012)*. Of course, this hypothesis remains fully compatible with the observation that the sediments from the Lujiatun Beds mostly represent a lahar (*Zhao, Barrett & Eberth, 2007*). It can be hypothesized that the burrows containing the *Changmiania* skeletons collapsed during the debris flow episode; we can alternatively imagine that the *Changmiania* specimens dug their burrow in unstable reworked volcanic material just after the debris flow. Those explanations of course remain pure speculations, as firsthand stratigraphic and taphonomic data are lacking for the currently known *Changmiania* specimens.

## CONCLUSIONS

*Changmiania liaoningensis* nov. gen., nov. sp., from the Lower Cretaceous Lujiatun Beds of western Liaoning Province (China), is placed at the base of the clade Ornithopoda. The perfect preservation of the skeleton of both its holotype and referred specimen in a lifelike posture implies that the animals were rapidly entombed while they were still alive. Although its hindlimb proportions indicate that *Changmiania* was an efficient cursorial dinosaur, some of is diagnostic features are tentatively interpreted as adaptations to a fossorial behavior. It is therefore tentatively hypothesized that both *Changmiania* specimens were suddenly entrapped in a collapsed underground burrow while they were resting.

## INSTITUTIONAL ABBREVIATIONS

**BYU**      Brigham Young University, Provo, Utah, USA
**IGM**      Institute of Geology of Mongolia, Ulanbaatar, Mongolia
**IVPP**     Institute of Vertebrate Paleontology and Paleoanthropology, Beijing, China

| KDRC | Korea Dinosaur Research Center, Chonnam National University, Gwanju, Republic of Korea |
| MOR | Museum of the Rockies, Bozeman, Montana, USA |
| NHMUK | Natural History Museum, London, UK |
| PMOL | Paleontological Museum of Liaoning, Shenyang, China |
| YPM | Yale Peabody Museum of Natural History, New Haven, Connecticut, USA |

## ACKNOWLEDGEMENTS

Thanks first and foremost to Sun Ge, Hu Dongyu, and the staff of the Paleontological museum of Liaoning for extensive access to the specimens over the years. D. Barta, E. Brown, R.J. Butler and L.J. Krumenacker provided thoughtful comments on a previous draft of this manuscript that greatly improved its quality.

### Funding

This work was supported by the Belgian Federal Science Policy (BR/143/A3/COLDCASE). The funders had no role in study design, data collection and analysis, decision to publish, or preparation of the manuscript.

### Grant Disclosures

The following grant information was disclosed by the authors:
Belgian Federal Science Policy: BR/143/A3/COLDCASE.

### Competing Interests

The authors declare that they have no competing interests.

### Author Contributions

- Yuqing Yang analyzed the data, prepared figures and/or tables, and approved the final draft.
- Wenhao Wu conceived and designed the experiments, prepared figures and/or tables, and approved the final draft.
- Paul-Emile Dieudonné performed the experiments, analyzed the data, authored or reviewed drafts of the paper, and approved the final draft.
- Pascal Godefroit conceived and designed the experiments, performed the experiments, analyzed the data, prepared figures and/or tables, authored or reviewed drafts of the paper, and approved the final draft.

### Data Availability

Both the holotype (accession number: PMOL AD00114) and referres specimen (accession number: PMOL LFV022) are deposited in the collections of the Paleontological Museum of Liaoning, Shenyang Normal University, Shenyang, Liaoning Provice, China.

The data is available in the Supplemental Files.

## New Species Registration

The following information was supplied regarding the registration of a newly described species:

Publication LSID: urn:lsid:zoobank.org:pub:9C5F2451-4E00-4919-9FE9-14E6629FCF64.

*Changmiania* LSID: urn:lsid:zoobank.org:act:AE88C0EE-9C4B-463B-B271-EB2868D7C81E.

*liaoningensis* LSID: urn:lsid:zoobank.org:act:0F980B4C-1EB3-4137-93FB-2C38008EE5E2.

## Supplemental Information

Supplemental information for this article can be found online at http://dx.doi.org/10.7717/peerj.9832#supplemental-information.

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
