# Peer review of "A new basal ornithopod dinosaur from the Lower Cretaceous of China"

_PeerJ, doi:10.7717/peerj.9832_

## Round 0.1 · original submission · Minor Revisions

Based on the reviewer comments, this manuscript provides a convincing case for creation of this new taxon. The description and figures are well done, and the conclusions are appropriate to the data at hand. The reviewers did suggest a number of relatively minor comments, with the most important highlighted below:

- Oryctodromeus is a key taxon here given its inferred lifestyle and relative phylogenetic proximity. The relevant Krumenacker Ph.D. dissertation and other recent papers should be incorporated as appropriate. Note that I am OK with citing a dissertation document, because it is publicly and widely available.
- As suggested by multiple reviewers, the hand morphology should be discussed/described in more detail, even if the region is incomplete.
- The cervical vertebral count should be discussed in a little more detail, along the lines of comments from the reviewers.
- Please include the TNT or NEXUS file also for the phylogenetic analysis, to ease analysis of the files by the reader.

·

Basic reporting

Some slight grammatical changes have been suggested in the "General Comments to the Author" for more clear English.

Experimental design

Excellently done and no comment.

Validity of the findings

Excellent and no comment.

Additional comments

Line 47: Change meters to meter.

Line 75: It would be beneficial to hear more details on what exactly the restorations entail. This would facilitate confidence in the anatomical description.

Line 93: walker should be walkeri

Line 131: A phylogenetic and osteological description of Oryctodromeus is in post-review revision now. Krumenacker, 2017, a PhD thesis on the osteology, phylogeny, taphonomy, and ontogenetic histology of Oryctodromeus may be useful and is available at: https://www.researchgate.net/publication/324165136_Osteology_phylogeny_taphonomy_and_ontogenetic_histology_of_Oryctodromeus_cubicularis_from_the_middle_Cretaceous_Albian-Cenomanian_of_Montana_and_Idaho

Line 231: Change weaklier to something like “more weakly”

Line 302: Change wedge to wedges

Line 314: Sentence fragment that should be combined with subsequent sentence.

Line 409: It is mentioned in reference to dentary tooth count that: …”in any case, more than 15 teeth were present”. While I don’t myself doubt this. I would suggest briefly mentioning why this is a reasonable hypothesis since the dentary teeth are not visile.

Line 422: “Dental” should be “dentary” and “le” should presumably be “the”

Line 475: I suggest rephrasing “there is apparently then” to something like “then there is an apparent…”

Line 503: Should “artefact” be spelled “artifact”?

Line 514: Do I understand correctly that the presence of seven sacral vertebrae is being estimated here only because of the fusion of neural spines? I am unsure how justified this may be. As mentioned below, a lateral view of the fused neural spines of the sacral vertebrae may help lend evidence to this.

Line 525: Insert “a” between “only” and “few”.

Line 555: Ossified tendons from the dorsal column through to the end of caudal column are also prominent in some specimens of Oryctodromeus. See Krumenacker, 2017 (link above).

Line 595: A study by Jamie Fearon, on the morphometrics of Oryctodromeus forelimbs and other ornithopods may be of use for you in this section: https://www.tandfonline.com/doi/abs/10.1080/02724634.2016.1078341

Line 708: It may help to have an arrow or other method to highlight the gastroliths in Figures 1A and 1B.

Line 711: Change “to” to “with”.

Line 738: thescelosaurid is miss-spelled

Line 742: “Form” should be “forms”.
Line 765: “display” should be “displays”.

Line 766: “Bird” should be “birds”.

Line 826: A recent paper on the taphonomic setting of Oryctodromeus specimens, and subsequent burrows, may be useful in your discussion: https://www.sciencedirect.com/science/article/abs/pii/S0031018219300835

Line 844: Short hands are mentioned. Can that be quantified/described/figured better in some way? The description mentions they are mostly eroded and covered.

Line 860: The dorsomedial inclination of the ilia- I have noticed this as well in small neornithischians. Could this be taphonomic. I myself think you are on to something, but perhaps justify why you think the orientation is not taphonomic in origin.

Line 862: Like the above comment, I would tighten up your justification for a seven centra sacrum if you can’t actually see the centra.

Line 879: I would suggest putting the word “possible” before fossorial. While your evidence and suggestions here are sound, they cannot be considered definitive.

Line 884: In reference to the use of the word “carcasses”, could it be argued the animals were buried alive?

Line 886: “They” should be “the”.

Figure 1: A formal scale bar in each image would be beneficial.

Figure 8c: Would it be possible to see a lateral view of the sacral neural spines to see the fusion better? It would help document the condition as described.

Overall I think this is a good contribution of a small new possibly fossorial ornithsichian. The authors have done well in citing the literature, describing the animal, interpreting taphonomic data, and justifying their conclusions.

·

Basic reporting

Basic reporting is generally good - minor edits and corrections on the language are provided below.

Experimental design

All methods and approaches are generally sound.

Validity of the findings

Taxonomic and phylogenetic results are supported by the data. Taphonomic and ecological interpretations are less well supported and somewhat speculative, but this is openly acknowledged by the authors.

Additional comments

This paper describes some exceptionally preserved and very significant ornithopod dinosaur specimens from the Early Cretaceous of China as a new species. The manuscript is generally sound, with a strong case being made for the taxonomic distinctiveness of the new species, a generally sufficiently detailed description, and an appropriate phylogenetic analysis. Our comments are generally minor, and primarily address minor typographical errors and requests for clarifications and in some cases additional detail.

One more substantial problem with the manuscript is the clade names applied to the phylogenetic topology in figures 11 and 12. The authors strictly use clade names as defined phylogenetically by previous authors, but because their recovered topology differs substantially from that of previous work, this leads to clades with very strange taxonomic contents. It is particularly difficult to countenance, for example, that the clade names Thescelosauridae and Thescelosaurinae (which are family and subfamily groupings in the Linnean system) should include within them all iguanodontians, including hadrosaurs and species like Iguanodon and Mantellisaurus. We make the strong recommendation that the authors do not use Thescelosauridae and Thescelosaurinae on their chosen topology – just avoid using these clade names (there is no reason that all previously defined clade names need to be applied to their tree).


Line 21: “subcomplete” is unclear. Maybe “nearly complete”?

Line 101: Misspelling of “Orodromeus”

Line 111: Misspelling of “makelai”

Lines 115–118: As above, best to avoid using Thescelosauridae and Thescelosaurinae in this paper.

Line 126: Full stop required.

Line 133: “In the present analysis, we have not coded the basal ornithopod Oryctodromeus cubicularis, from the middle Cretaceous Blackleaf Formation of southwestern Montana and the Wayan Formation of southeastern Idaho, USA (Varricchio et al., 2007; Krumenacker, 2010), pending the formal description of the numerous partial skeletons from the Wayan Formation (Krumenacker, 2010).” - PhD thesis from Krumenacker (2017) has a description of nearly complete Oryctodromeus with material from Wayan Fm.

Line 137: “run TNT” > “run in/using TNT”

Line 139: Perhaps the authors could clarify why some of the characters (#110, #149, #152, #205, #208) were ordered in Dieudonne et al. (2016) but not in this study?

Line 143: A number of institutional abbreviations are used in the tables that are not defined here. Also, the correct abbreviation for the Natural History Museum, London, is NHMUK.

Line 169: We’re confused about the use of “JMOL” as the prefix to both specimens of Changmiania when PMOL is stated in the institutional abbreviations on Line 143.

Line 200: Isn’t fusion of the premaxillae also present in Changchunsaurus?

Line 214: Perhaps the authors could state whether or not it is possible to determine if the rostral end of the premaxilla was edentulous from both specimens.

Line 225: “NHM” should be “NHMUK” throughout.

Line 230: “progressively weaklier expressed” > “progressively less expressed”?

Line 232: “significantly lesser than” > “significantly less than”

Line 242: “resent” should be “present”

Line 243: “up to a weak basal cingulum” is confusing as the cingulum is at the base of the crown – “down to a weak basal cingulum” would be more appropriate.

Line 247: “on the labial” should be “on the labial surface”

Line 250: “and inverted” should be “an inverted”

Line 251: “The dorsal portion of the ventral process is slightly longer rostrocaudally than its ventral part.” > maybe “The ventral process becomes more rostrocaudally restricted ventrally?”

Lin 275: Consistent use of either palpebral/supraorbital. “Palpebral” is used by the authors in-text, but “supraorbital” is used in the figures.

Line 276: “The caudal process of the palpebral is robust, perfectly straight” Maybe the authors should remove “perfectly”. The palpebral does look straight in dorsal view but it looks a bit curved in lateral view.

Line 277: remove comma between “particularly” and “long”.

Line 279: remove comma after “postpalpebral”

Line 281: “a wide visor” – “visor” is a rather odd descriptive term and difficult to understand exactly what the authors mean.

Line 284: “craniocaudally”. Perhaps the authors should use “rostrocaudally” for consistency?

Line 314: “borer” > “border”

Line 325: “its rostrocaudal end remains equal along its whole height.” We may have misunderstood this but do the authors mean “rostrocaudal width/length”?

Line 332: Another closed bracket required after “(e.g., Archaeoceratops (IVPP V11114)”

Line 332: What is Butler & Han (2009)? As far as we know this paper doesn’t exist!

Line 341: “it largely overlaps the squamosal process of the postorbital, reaching the postorbital process of the latter (Fig. 6B).” We don’t think the latter part is necessary so perhaps just “it largely overlaps the squamosal process of the postorbital (Fig. 6B).”

Line 344: “which forms the forms the” includes some unnecessary words.

Line 386: “appears” > “appear”

Line 388: “Changchunsosaurus” > “Changchunsaurus”

Line 411: don’t need “relative” at the start of this sentence because “relatively” is used later

Line 422: “dental teeth” > “dentary teeth”

Line 422: “le” > “the”?

Line 447: “still complicates” > “further complicates”

Line 449: The six cervical vertebrae is an absolutely remarkable character. Are the authors certain there is no reconstruction or modification of the specimens in this region? Is the count of six verifiable in both specimens?

Line 457: For Scelidosaurus, the authors should use the recent paper by David Norman in Zoological Journal of the Linnean Society. Eight cervical vertebrae are present in Scelidosaurus. Nine were present in Scutellosaurus, as reported by Colbert (1981).

Line 503: It’s unclear what you mean by “it can also be an artefact of preservation”.

Line 553: “anterior dorsals vertebrae” > “anterior dorsal vertebrae”

Line 598: Kulindadromeus should be italicised.

Line 620: full stop at the end of the sentence.

Line 620: Maybe the authors could expand on the description of the hands here. From Figure 1 the hands seem much better preserved in the referred specimen than the holotype.

Line 655: Butler (2005) argued for an elongate ischial symphysis in Lesothosaurus, although this is reduced through ontogeny if the synonymy of Stormbergia and Lesothosaurus is correct.

Line 655: Please mention that the pubis is not visible.

Line 669: Author don’t mention that the 4th trochanter is covered by sediment (I assume) in both specimens.

Line 671: “referred specimens”. There’s only one referred specimen so it should be “referred specimen” throughout?

Line 678: It is unclear what “it is more salient distally” means.

Line 685: “expended” > “expanded”

Line 690: “cranial view” should that be dorsal view? Also Figure 10D description.

Line 691: “Left digit I is partly exposed.” I can’t see it on holotype or Figure 10D – but it does seem to be partially visible on the right foot of the referred specimen. Maybe the authors could expand on it a bit more and talk about the digit 1 ungual here or around Line 703.

Line 738: “thescelosautid” misspelt “thescelosaurid”

Line 742: “Jeholosauridae” > “Jeholosaurinae”

Line 747: Remove comma after 75 degrees

Line 752: Polarity is misspelled

Line 771: “characteristic” > “characteristics”

Line 794: “Cruelly” is a bit dramatic and could just be cut.

Line 805: “peaceful lifestyle posture” > “peaceful posture”

Line 815: Pompei is double ‘ii’

Line 831: “Oryctodromeus is also an orodromine ornithopod (Varricchio et al., 2007; Boyd, 2015)” Phylogeny from Boyd (2015) doesn’t show orodromines as ornithopods so maybe remove “ornithopod” or Boyd’s citation.

Line 844: “short hands” this feature isn’t mentioned in the description of carpals & manus on Line 619.

Line 863: “larger Iguanodontia” > “larger iguanodontians/ids”?

Figure 1: Clearer scale required.

Figure 3: In the line drawing 3B it looks like there is articulation between the supraorbital and postorbital.

Figure 3 & 4: The label for supraoccipital is missing.

Figure 8 caption: “vie” > “view”

Figure 11 caption: “belo” > “below”

Table 2 caption: Some of the taxa in the Table are missing from the caption. Also, some some of these abbreviations have been missed out of the in-text institutional abbreviations on Line 143. Also “NHM” > “NHMUK” throughout.

Supplementary info: Perhaps the authors could include a TNT file or nexus file in addition to the matrix provided to make it easier to analysis the tree.

Supplement 1: Character #110 is described as ordered in the character list whereas in-text the authors state that all characters were unordered.


Emily Brown & Richard Butler (co-signed review)

·

Basic reporting

Line 231: “progressively weaklier” should be “more weakly.”
Line 242: Typo: “resent” should be “present.”
Line 250: Typo: “and” should be “an.”
Line 314: “borer” should be “border.”
Line 326: “its rostrocaudal end remains equal along its whole height.” Please clarify which specific dimension remains equal in this case? Width? In which anatomical direction?
Line 386: Typo: “appears” should be “appear.”
Line 422: Typos: “Dental” should be “dentary” and “le” should be “the.”
Line 529: “At all” should be “Overall.”
Line 580: Typo: “Makovicky et ala.” should be “Makovicky et al.”
Line 691: Typo: “tibiotasus” should be “tibiotarsus.”
Line 692: Typo: “formed” should be deleted.
Line 738: Typo: “thescelosautid” should be “thescelosaurid.”
Line 750: To which element does the “shaft” belong?
Line 771: Typo: “characteristic” should be “characteristics.”
Line 866: Typo: “appear” should be “appears.”
Line 886: Typo: “they” should be “their.”


Figures: A larger close-up photo of a maxillary tooth is needed to more easily compare the teeth of the new specimens to those of other taxa. It is difficult to see the apicobasal ridges that are compared to those of Jeholosaurus, Changchunsaurus, and Hypsilophodon.

A figure of one of the skulls in occipital or a partial occipital view should be included to illustrate the supraoccipital and any other visible elements.

Experimental design

Lines 69-81: The results of the X-ray analyses and photographs of the specimens during preparation should be included in the supplementary information for the sake of transparency and reproducibility of the results.

Lines 88-121: Madzia et al. (2018) provide an updated list of clade names and definitions in their supplementary information. Please make sure that your usage of names and definitions conforms to the new names and rationales that they provide.

Line 137: To enhance the repeatability of their work, the authors should consider uploading their .tnt file with the list of commands they used for the phylogenetic analysis.

Validity of the findings

Line 173: laterally expanded nasal: How does this differ from the condition in Haya griva, in which the nasal also hangs over the maxilla slightly (Makovicky et al., 2011; Norell and Barta, 2016)?
Line 174: Please use “supraorbital” instead of “palpebral” here to be consistent with the rest of the manuscript and the recommendations of Maidment and Porro (2010) and Nesbitt et al. (2012).
Line 435: The shape of the ventral margin of the angular should be compared to that of Haya. There is a lesser degree of curvature present in Haya (Makovicky et al., 2011; Norell and Barta, 2016). So, it may simply be the degree of curvature of the ventral border of the angular that is autapomorphic for the new taxon, not the presence or absence of this curvature itself. In general, comparisons of the surangular, angular and articular to those of other taxa would enhance the usefulness of the descriptions of these elements.
Line 449: The shortened neck composed of six cervical vertebrae is a clear autapomorphy among basal ornithopods. An additional figure of one of the articulated necks in either dorsal or lateral view would more clearly illustrate this character.
Line 533: The proximal caudals are not visible in Fig. 8B. Please include a close-up figure of these vertebrae.
Line 619: The authors should provide close-up figures of the manus and carpus elements. Hands and wrists are rarely preserved among basal ornithopods, and even disarticulated elements may be identifiable and informative (e.g., Norell and Barta, 2016).
Lines 657-706: Additional comparisons of the femur, tibia, fibula, and pes to those of other taxa would be very welcome. The fusion of the astragalus and calcaneum to the tibia and the robustness of the fibula are particularly noteworthy characters that should be discussed in a comparative context.

The postorbital and squamosal region of the new specimens is highly unusual for a basal neornithischian/ornithopod, supporting the authors’ contention that it represents a distinct genus and species.

The shapes of the new taxon’s skull elements do not easily fit within or extrapolate from trends observed from the partial ontogenetic sequence of Jeholosaurus (Barrett and Han, 2009) known to date, supporting the authors’ claim that the two taxa are distinct.

It is not clear from the description whether or not all of the observations refer to the holotype skull or to both skulls. The authors should note any variation present between the two skulls they examined, or provide a general statement stating that they’re congruent in all respects. Though the sample size is small, it’s important that future workers be aware of any individual variation (or lack thereof) within the new taxon.

Additional comments

The authors describe two new specimens of a heretofore unknown taxon of basal ornithopod dinosaur from the Early Cretaceous of northeastern China. My major concern upon first receiving the manuscript was that these specimens would not prove to be distinct from the contemporaneous Jeholosaurus; however, the authors have allayed this concern of mine through their careful comparisons and phylogenetic analysis. The new specimens have several unusual diagnostic features of their cranial and postcranial anatomy, notably the shapes and arrangement of the bones surrounding the infratemporal fenestra. Postcranially, the shortened neck composed of six cervical vertebrae is a clear autapomorphy among basal ornithopods.

Overall, the description of the specimens is well-written and requires few changes. Where changes are warranted, I have indicated these in the “Basic Reporting,” “Experimental Design,” and “Validity of Findings” sections. The manuscript would benefit from a few additional figures, noted in each section.

The phylogenetic analysis appears to have been conducted appropriately, and the results are reasonable given the matrix used. The authors might note that Boyd (2015) and other derivations of that matrix do not recover Jeholosaurinae as a clade. Of course, the position of the majority of the taxa discussed as either inside or outside Ornithopoda is an ongoing debate. The authors should briefly discuss why they think their matrix recovers the majority of the taxa as ornithopods, in contrast to Boyd (2015).

The discussion of anatomical characteristics supporting a facultative burrowing lifestyle is well supported with comparisons to both extant taxa and Oryctodromeus and related forms for which many of the same burrowing-related features were previously proposed. The authors appropriately label their interpretations of the taphonomy as speculations, given that most of the taphonomic context of these specimens has been lost.

I congratulate the authors on an important addition to knowledge of basal neornithsichian/ornithopod anatomy and diversity.

Sincerely,
Daniel Barta

Literature Cited:
Barrett PM, Han F-L. 2009. Cranial anatomy of Jeholosaurus shangyuanensis (Dinosauria: Ornithischia) from the Early Cretaceous of China. Zootaxa 2072:31–55.
Boyd CA. 2015. The systematic relationships and biogeographic history of ornithischian dinosaurs. PeerJ 3:e1523. DOI: 10.7717/peerj.1523.
Madzia D, Boyd CA, Mazuch M. 2018. A basal ornithopod dinosaur from the Cenomanian of the Czech Republic. Journal of Systematic Palaeontology 16:967–979. DOI: 10.1080/14772019.2017.1371258.
Maidment SCR, Porro LB. 2010. Homology of the palpebral and origin of supraorbital ossifications in ornithischian dinosaurs. Lethaia 43:95–111. DOI: 10.1111/j.1502-3931.2009.00172.x.
Makovicky PJ, Kilbourne BM, Sadleir RW, Norell MA. 2011. A New Basal Ornithopod (Dinosauria, Ornithischia) from the Late Cretaceous of Mongolia. Journal of Vertebrate Paleontology 31:626–640. DOI: 10.1080/02724634.2011.557114.
Nesbitt SJ, Turner AH, Weinbaum JC. 2012. A survey of skeletal elements in the orbit of Pseudosuchia and the origin of the crocodylian palpebral. Earth and Environmental Science Transactions of the Royal Society of Edinburgh 103:365–381. DOI: 10.1017/S1755691013000224.
Norell MA, Barta DE. 2016. A New Specimen of the Ornithischian Dinosaur Haya griva, Cross-Gobi Geologic Correlation, and the Age of The Zos Canyon Beds. American Museum Novitates 3851:1–19.

---

## Round 0.2 · accepted · Accept

Thank you for your close attention to the comments from the reviewers. In my opinion, the manuscript is ready to proceed to publication.

During the proof phase, you may wish to correct the following minor typographic errors/stylistic suggestions:
line 75 -- replace "too fragilized" with "overly fragile" or simply "fragile"
line 229 -- add comma in sentence "They do not bear denticles, and their "
line 776: remove space before colon, so it reads "ornithopods: the frontals are"